# Gap-filled sub-surface mooring dataset off Western Australia during 2010–2023

**Toan Bui[1], Ming Feng[1], and Christopher C. Chapman[2]**

[1]CSIRO Environment, Indian Ocean Marine Research Centre, Crawley, WA, Australia
[2]CSIRO Environment, Hobart, TAS, Australia

**Correspondence:** Ming Feng (ming.feng@csiro.au)

**Abstract.** Coastal moorings allow scientists to collect long-term datasets that are valuable in understanding shelf dynamics, detecting climate variability and changes, and evaluating shelf dynamics and climate variability and change impacts on marine ecosystems. However, we often cannot obtain continuous time series data from moorings due to mooring losses or instrument failures. Here, we present an updated version of the 14-year sub-surface mooring dataset off the south-western coast of Western Australia (WA) during 2010–2023 (https://doi.org/10.25919/myac-yx60, Bui and Feng, 2024). This updated dataset offers continuous daily temperature and current data with a 5 m vertical resolution, collected from six coastal Integrated Marine Observing System (IMOS) moorings at depths between 48 and 500 m. The self-organizing map (SOM) machine learning technique is applied to fill in the data gaps in the previous version. The data capture the Leeuwin Current variability on the shelf from intraseasonal to interannual timescales. The data also capture the variability of the Capes Current, a wind-driven northward current on the middle shelf. The usage of the in-filled data product is demonstrated by detecting extreme temperature events on the Rottnest Shelf. The data products can be used to characterize sub-surface features of extreme events such as marine heat waves and marine cold spells, which are influenced by the Leeuwin Current and the wind-driven Capes Current, and to detect decadal change signals along the WA coast.

## 1 Introduction

Oceanography moorings are underwater instruments anchored on the sea floor that collect ocean currents, temperature, salinity, and other environmental parameters. Typically, mooring deployment periods range from 4 to 6 months in shelf waters to up to 18 months in deep oceans (Sloyan et al., 2024). Sustained long-term mooring observations serve as invaluable resources for environmental and climate research and play a vital role in calibrating and validating numerical models (Bailey et al., 2019).

The south-western Western Australia (WA) mooring array is part of the Integrated Marine Observing System (IMOS) program operated by the Commonwealth Scientific and Industrial Research Organisation (CSIRO) since 2009, designed to monitor the influences of the southward-flowing Leeuwin Current (LC) on the continental shelf (Thompson, 1984; Chen and Feng, 2021). The anomalous meridional pressure gradient, associated with warm, low-salinity waters from the tropical Pacific Ocean entering the Indian Ocean through the Indonesian Archipelago, is the main driver of the LC (Feng and Wijffels, 2002; Godfrey and Ridgway, 1985). The strength of the LC varies seasonally, mostly due to variations in the along-shore winds (Smith et al., 1991). During the austral summer, strong along-shore northward winds drive the Capes Current northward on the central inner shelf (Fig. 1). The interannual variability of the LC is often associated with remote signals from the Pacific, i.e. the El Niño–Southern Oscillation (ENSO), the current being stronger during La Niña and weaker during El Niño (Feng et al., 2003).

The south-western WA mooring array has helped scientists identify the key role of the LC in the development of marine

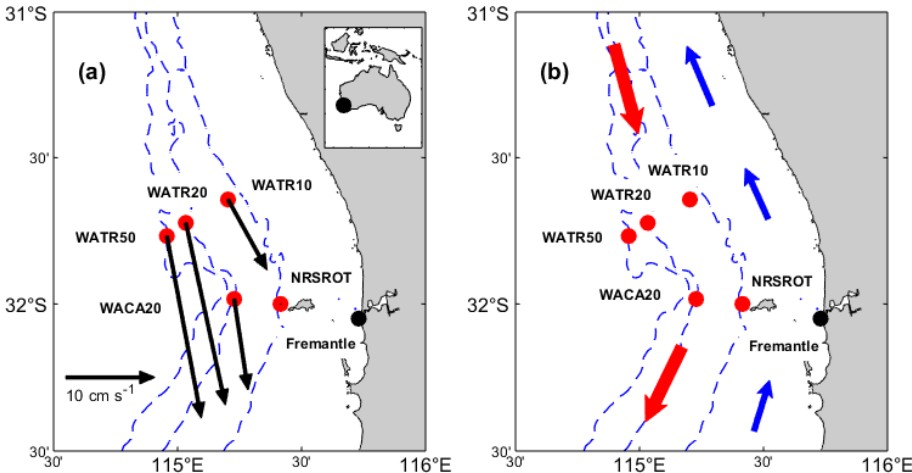

**Figure 1.** Bathymetry map and mooring locations (red circles) on the Rottnest Shelf. **(a)** Velocities estimated from measurements, with black arrows representing the mean state of vertically averaged velocities. The 0–200 m average is used for the WATR50 mooring. The three dashed lines represent the 50, 200, and 500 m contours. The black circles indicate the location of the Fremantle tide gauge station. Note that NRSROT consists of two separate moorings. **(b)** Schematic of the shelf currents, with the red arrows denoting the Leeuwin Current and the blue arrows indicating the direction of the wind-driven Capes Current.

heat waves (MHWs) off the coast (Benthuysen et al., 2014; Feng et al., 2013). The mooring data were also employed by Feng et al. (2021) to detect abnormal cooling events off the coast over 2016–2019 (defined as the marine cold spell or MCS), when the thermocline depth was elevated due to the weakening of the LC during the El Niño events. The sustained IMOS mooring array encompasses six coastal moorings on the Rottnest Shelf during 2010–2023 that ranged from 50 to 500 m (Fig. 1 and Table 1). The first version of the gridded data from these moorings was published by Chen and Feng (2021), and an extension was published by Bui et al. (2023). Mooring time series are susceptible to missing values due to mooring loss and instrument failure. Strong currents can exert a force on the mooring line, causing it to be pushed down into the water column and leaving data gaps near the surface (Sloyan et al., 2023). This paper introduces a new update of the mooring data, filling data gaps using a statistical method.

Various techniques have been employed to address gaps in mooring datasets. Sprintall et al. (2009) utilized a damped least-squares fitting method to fill substantial gaps in mooring current time series data when estimating the Indonesian throughflow transport. Wang et al. (2015) adopted a combination of data extrapolation, data interpolation, and a least-squares regression model to fill in missing data recorded in the central equatorial Indian Ocean. Cao et al. (2015) employed harmonic analysis and modal decomposition to isolate the tidal currents for each mode and reconstruct the full-depth tidal currents in the northern South China Sea. More recently, Sloyan et al. (2023) experimented with a machine learning approach, a self-organizing map (SOM), to fill data gaps in the East Australian Current mooring array. The

choice of method depends on the characteristics of data loss, such as the duration of gaps or the depth range affected, as well as the intended analyses of the data.

A SOM is a technique that projects high-dimensional input data onto a two-dimensional output space while preserving the topological structure of the input data (Kohonen, 1982). In a SOM, units are organized so that similar units are positioned close to each other, while dissimilar ones are separated in the output data space. This method has found extensive applications in meteorology and oceanography (Liu and Weisberg, 2011) and can perform a range of tasks, including clustering, data analysis and visualization, feature extraction, and data interpolation (Lobo, 2009).

Chapman and Charantonis (2017) utilized a SOM to reconstruct deep current velocities in the Southern Ocean from sea surface observations. They used densely observed surface velocities, sea surface height, sea surface temperature from satellites, and sparsely observed deep current velocities from Argo floats to train the SOM. Then, they derived dense velocity fields at a depth of 1000 m. Their method took advantage of local correlations in the data space to find the smallest Euclidean distance, weighted by the local correlations, between a vector with missing components in the data space and the SOM units, which increased the accuracy of the filled deep velocities.

This study employs the SOM method to fill in the data gaps in the south-western WA mooring data, following the procedure in Chapman and Charantonis (2017), to generate a gap-free time series dataset. The use of the continuous dataset is demonstrated by examining several extreme temperature events that occurred in the region.

**Table 1.** Summary of the coastal mooring stations. NRSROT: National Reference Station west of Rottnest Island. WACA: Western Australia Perth Canyon. WATR: Western Australia Two Rocks.

| Station | Latitude; longitude | Station depth (m) | Temperature | | | | ADCP | | | |
|---|---|---|---|---|---|---|---|---|---|---|
| | | | Instrument | Interval (min) | Mean sensor depths (m) | Data span | Instrument | Interval (min) | Bin numbers × bin size | Data span |
| NRSROT-Temperature | 31.9900° S; 115.3850° E | 61 | SBE39[a]SBE37[b] | 5–15 | 27; 33; 43; 55 | Jan 2010–May 2023 | | | | |
| NRSROT-ADCP | 32.0000° S; 115.4170° E; | 48 | | | | | RDI Workhorse 600 kHz[c]; Nortek Signature 500 kHz[d] | 15 | 11 × 4 m | Aug 2011–May 2023 |
| WACA20 | 31.9830° S; 115.2280° E | 200 | | | | | Nortek Signature 250 kHz[d]; Nortek Continental 190 kHz[d] | 15 | 41 × 5 m | Aug 2011–May 2023 |
| WATR10 | 31.6433° S; 115.2033° E | 100 | SBE39 SBE37 | 5–15 | 25; 30; 35; 40; 52; 70; 90 | Jan 2010–May 2023 | Nortek Aquadopp 400 kHz[d]; | 15 | 17 × 5 m | Aug 2011–May 2023 |
| WATR20 | 31.7233° S; 115.0333° E | 200 | SBE39 SBE37 | 5–15 | 25; 35; 50; 68; 100; 125; 150; 175 | Jan 2010–May 2023 | Nortek Continental 190 kHz[d]; Nortek Signature 250 kHz[d] | 15 | 25 × 8 m | Aug 2011–May 2023 |
| WATR50 | 31.7683° S; 114.9567° E | 500 | | | | | RDI Long Ranger 75 kHz[c]; Nortek Signature 55 kHz[d] | 15 | 26 × 20 m | Aug 2011–May 2023 |

[a] SBE39 (and SBE39 plus) is a self-contained, autonomous temperature logger (SBE: Sea-Bird Electronics). [b] SBE37 is a single-channel CTD (conductivity–temperature–depth) sensor. [c] RDI ADCPs (acoustic Doppler current profilers) are manufactured by Teledyne RD Instruments and comprise Long Ranger 75 kHz and Workhorse 600 kHz (https://www.teledynemarine.com/rdi, last access: 9 April 2025). [d] Nortek ADCPs are produced by the Nortek group, including Signature 55 kHz, Continental 190 kHz, Signature 250 kHz, Aquadopp 400 kHz, and Signature 500 kHz (https://www.nortekgroup.com, last access: 9 April 2025).

## 2  Data and methods

### 2.1  Moored instrument data

#### 2.1.1  Temperature

The in situ temperature dataset is collected using Sea-Bird Electronics instruments, including SBE37, SBE39, and SBE39 plus, with sampling intervals varying between 5 and 15 min (Table 1). To ensure data quality, the raw dataset underwent rigorous quality assurance (QA) and quality control (QC) procedures (Morello et al., 2014), utilizing the IMOS mooring toolbox written in the MATLAB scientific programming language. Only data flagged as 1, indicating good quality, are retained for this analysis. The QC data are concatenated and then linearly interpolated onto a grid of 5 m vertical resolution, and they are averaged daily (Bui et al., 2023). The unfilled data are available in the CSIRO Data Access Portal (https://doi.org/10.25919/9gb1-ne81, Bui et al., 2023).

For data completion, we use satellite sea surface temperature (SST) sourced from the Regional Australian Multi-Sensor SST Analysis (RAMSSA) version 1.0 (Beggs et al., 2011) to extend the temperature data at each mooring to the sea surface by linear interpolation. The RAMSSA system combines SST data from infrared and microwave sensors on polar-orbiting satellites with in situ measurements to generate daily foundation SST. North of 40° S, RAMSSA is on average within ±0.07 °C of other multi-sensor SST analyses (Beggs et al., 2011). From conductivity–temperature–depth (CTD) profiles in the study region, ocean temperatures vary mostly linearly in the near-surface layer (top 30 m, below the foundation SST depth), so linear interpolation is an acceptable approximation.

When minor gaps occur near the bottom, we use two available data points at the bottom of the vertical temperature profile to extrapolate linearly to the sea bottom.

These procedures produce gridded temperatures with daily 5 m vertical resolution at the NRSROT, WATR10, and WATR20 moorings, spanning the time period from January 2010 to May 2023, as presented in Fig. S1 in the Supplement.

#### 2.1.2  Velocity

The velocity observations on the IMOS mooring array are recorded by various RDI and Nortek ADCP instruments, typically sampling at 15 min intervals and mounted in an upward-looking configuration above the seabed (Table 1).

The raw velocity data undergo quality control procedures similar to temperature, followed by concatenation and interpolation into a daily grid with 5 m vertical resolution, as described by Bui et al. (2023). The velocity dataset comprises observations from five stations: NRSROT, WACA20, WATR10, WATR20, and WATR50. Initially, gaps in the time series are filled using linear interpolation if the temporal gap size is less than 3 d. Subsequently, for each velocity profile, gaps near the surface or bottom are filled using linear

extrapolation, akin to the technique applied for temperature data. The meridional and zonal components of the velocity datasets, from August 2011 to May 2023, are presented in Figs. S2 and S3, respectively.

For the 2010–2023 period, the percentage of missing mooring data varies from 2 % to 16 % for temperature and from 12 % to 33 % for velocity at various moorings (Table 2). The largest percentage of missing data is at WATR20, which is situated in the core of the LC system. The percentages of missing data tend to have high values near the surface and bottom layers of a mooring due to the mooring movement and variations of deployment depth over time (Fig. S4).

### 2.2  SOM method

To produce a gap-filled data product, we follow the method described in Chapman and Charantonis (2017). As discussed briefly in the introduction, this method "completes" a gappy dataset by first using available data to train a SOM, which effectively clusters the data into a set of discrete states. These states can be represented as a two-dimensional map, where neighbouring clusters are more similar to each other than distant clusters. Associated with each cluster is a *referent vector* that approximates the mean of all data assigned to that cluster and a weighted mean of data assigned to neighbouring clusters. After the map is trained, new data can be assigned to existing clusters by comparing the Euclidean distance in the data space between that new data vector and the referent vector of each cluster. The cluster with the smallest Euclidean distance is known as the best-matching unit (BMU). Once a SOM is available, data vectors with missing components are presented sequentially, the BMU is found, and the missing data are completed (in-filled) by replacing them with the relevant components of the referent vector of the BMU. For full details, see Chapman and Charantonis (2017).

A schematic using the SOM method to fill gaps in the mooring dataset is shown in Fig. 2. We utilized the Vesanto et al. (2000) SOM toolbox for MATLAB 5 in this study. The temperature or velocity data for each station, along with ancillary data, are aggregated into data matrices. The ancillary data include the day of the year and the daily Fremantle sea level (Fig. 1). Sea level data are obtained from the University of Hawaii's Sea Level Center (https://uhslc.soest.hawaii.edu/, last access: 9 April 2025). The Fremantle sea level serves as a proxy for the annual and interannual variations of the Leeuwin Current (Feng et al., 2003). We have tested adding along-shore winds to the data matrices. However, there is no improvement in the results, so wind data are not used in the SOM calculation, as the wind information may have been integrated into the sea level data. The temperature input matrix comprises 4869 rows (representing the number of time steps) and 77 columns (reflecting the number of different observations at each time step). Similarly, the velocity input matrix consists of 4292 rows and 361 columns. The temperature–velocity input matrix with missing values

**Table 2.** Percentage (%) of missing temperature and velocity for each mooring for the time period 2010–2023. Note that temperature profiles are not available at WACA20 and WATR50. NA – not available

|          | Temperature (%) | Velocity (%) |
|----------|:---------------:|:------------:|
| NRSROT   | 2               | 12           |
| WACA20   | NA              | 19           |
| WATR10   | 7               | 18           |
| WATR20   | 16              | 33           |
| WATR50   | NA              | 21           |

is indicated by Dataset 1 in Fig. 2. Only fully available profiles in the input matrix are selected as the training data in Dataset 2. Consequently, the number of rows in the training data is 3675 for temperature and 1146 for velocity.

The number of units in the SOM is specified prior to the training process. According to the literature, a small number of SOM units is useful in capturing the general features of the system (Liu and Weisberg, 2011), while a larger number provides more detailed information and is more suitable for data gap filling (Sloyan et al., 2023). In our case, where we aimed to capture detailed information from the training data containing a large number of profiles, we opted for a larger number of units, i.e. 1000 units for the temperature data and 500 units for the velocity data. Using lower numbers of units only had minor effects on the results. We used a batch algorithm to train the SOM (Chapman and Charantonis, 2017). The training phase of the SOM was done in two steps: a first rough phase and a fine-tuning phase. In the first step, the neighbourhood radius and learning rate were set to some high values in order to gain a general orientation of the map, while in the second step they were set to smaller values to make only fine adjustments to the SOM unit's position.

One of the important steps was the assignment of each input vector to a specific SOM unit, $u$, shown on the right-hand side of Fig. 2. Firstly, we estimated the local correlations in the data space, represented by a $\mathrm{cor}_{i,j}^u$ matrix:

$$\mathrm{cor}_{i,j}^u = 1 + \sqrt{\sum \mathrm{DAT\_cor}^2}, \qquad (1)$$

TS1 where DAT_cor is a correlation matrix for each normalized input vector within a SOM unit. $\mathrm{cor}_{i,j}^u$ is the local correlation matrix between the missing data and the mean of all the observed training data within the SOM unit $u$.

Given with local correlations in the data space, we then calculated the minimum Euclidean distance between a normalized input vector $X$ containing missing and non-missing components and the referent vector of the SOM unit, $\mathrm{ref}^u$, using a similarity function (Chapman and Charantonis, 2017).

The similarity function is defined as

$$\mathrm{sim}\left(X, ref^u\right) = \sum_{i \in \text{non-missing}} \left(1 + \sum_{j \in \text{missing}} \left(\mathrm{cor}_{i,j}^u\right)^2\right) \\ \times \sqrt{\left(X_i - ref_i^u\right)^2}, \qquad (2)$$

where $X_i$ is the non-missing data in $X$ and $ref_i^u$ is the mean of all training data in the SOM unit $u$. After determining the most appropriate SOM unit, the missing values in the input vector were extracted from the corresponding referent vector, providing the in-filled data in Dataset 3 (Fig. 2).

## 2.3 Validation of the SOM-based infilling technique

For mooring data, a failed mooring or instrument often results in a block of data being lost until the next deployment. To simulate this effect, we withhold temperature data at one site for 150 d from 1 January to 30 May 2020, which is roughly the length of one deployment cycle. We utilize temperature data at the other two sites to identify the best-matching SOM units and to fill in the withholding data. At NRSROT, the $R^2$ and the root mean square error (RMSE) between withheld and filled temperature data are 0.70 and 0.61 °C, respectively. At WATR10, these values are 0.86 and 0.39 °C, and at WATR20 they are 0.91 and 0.58 °C, as shown in Fig. 3. Furthermore, we evaluate the ability of the SOM method to reconstruct extreme temperature patterns. As shown in Fig. S5, a comparison of the observed and SOM-derived temperatures at WATR20 during the validation period (1 January to 30 May 2020) highlights this ability. The black crosses in both panels denote days identified as marine cold spells, which are defined as periods where temperatures fall below the 10th percentile for at least 5 consecutive days (Hobday et al., 2016). SOM-derived temperatures successfully captured three bottom-intensified MCS events as in the observations, demonstrating the method's reliability in reconstructing extreme cold temperature patterns.

To assess potential overfitting, the SOM method was tested on a separate period spanning the time from 10 January to 8 June 2012, with 150 d withheld from training. The resulting RMSE values were 0.41 °C at NRSROT, 0.36 °C at WATR10, and 0.55 °C at WATR20. If we repeat this process and validate the method against the data included in the training dataset, we obtain RMSE figures similar to those obtained from the withheld data, indicating that the SOM method does not overfit the dataset.

To assess the accuracy of the SOM method further, we compare it with a simple climatology method over the same validation period, as shown in Fig. S6. Overall, the mean vertical temperature profiles from the SOM method are closer to the observed data than those from the climatology method (Fig. S6a–c). As a result, the residuals from the SOM method, calculated by subtracting the filled SOM val-

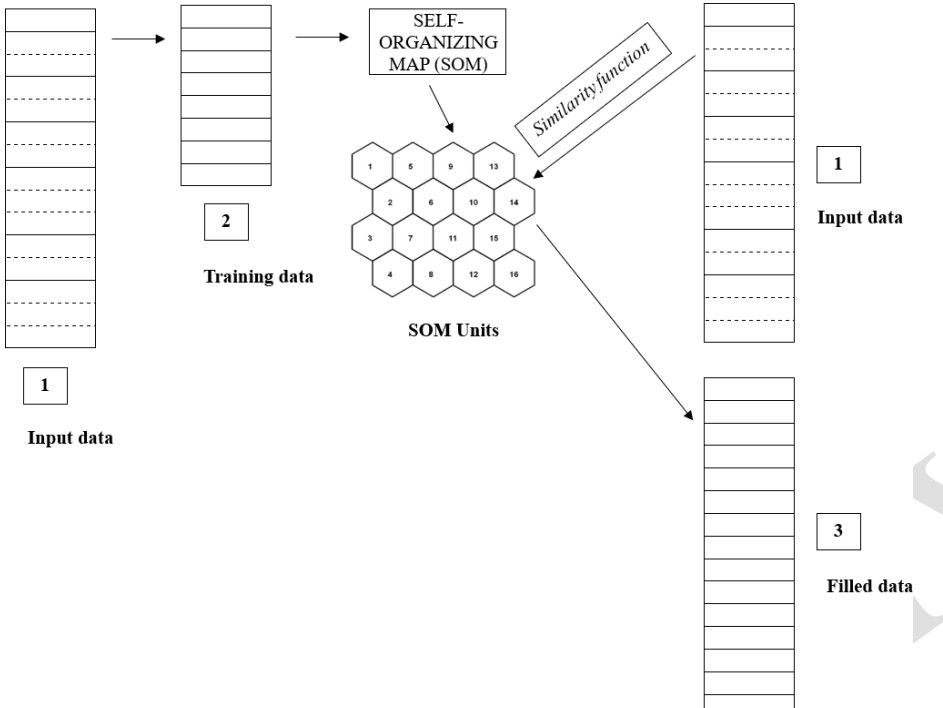

**Figure 2.** Schematic of the SOM method applied to fill gaps in the mooring temperature and velocity data. Dataset 1 denotes an input data matrix in which rows are daily time vectors and columns are observational variables. In Dataset 1, solid lines present full available profiles, while dashed lines show the profiles including missing values. In Dataset 2, only full data profiles are selected for training a SOM. In a SOM, we pre-define the number of units, e.g. 1000 units for temperature and 500 units for velocity. Each SOM unit contains a referent vector. On the right-hand side, each daily input vector in the input data matrix is assigned to each SOM unit using a similarity function defined by Chapman and Charantonis (2017). Finally, we use the referent vector of each SOM to fill gaps in the corresponding daily input vector, as shown in Dataset 3.

ues from the observed temperatures, are smaller than the climatological residuals. Additionally, the standard deviation of the observed temperatures is closer to that of the SOM data, while it differs significantly from that of the climatological values (Fig. S6d–f). These findings suggest that the SOM method is more reliable than the climatology method.

Using the same approach, we examine the accuracy of velocity data gap filling. Specifically, we consider the period from May to August 2020, during which velocity data at WATR50 within the depth range of 70–450 m are withheld for 90 d. For the meridional velocity, the $R^2$ and RMSE values between the withheld and in-filled data are 0.63 and $0.12 \, \mathrm{m \, s^{-1}}$, respectively (Fig. 4a). For the zonal component, these values are 0.50 and $0.05 \, \mathrm{m \, s^{-1}}$, respectively (Fig. 4b). To determine whether the SOM method overfits the data, we withheld velocity data from a different period spanning the time from May to August 2012. The resulting RMSE values for the meridional and zonal velocities are 0.13 and $0.06 \, \mathrm{m \, s^{-1}}$, respectively. These findings align with the RMSE from the validation data, indicating that the SOM method effectively avoids overfitting.

## 3   Data application

Having confirmed the effectiveness of the SOM method in filling missing values in a mooring dataset, we now employ all non-missing daily data to train the SOM and then fill the data gaps. The filled temperature data exhibit consistent temporal and spatial variability (Fig. 5). The gap-filled data capture cold temperature events at WATR20 during early 2010 and mid-2016, coinciding with periods when the thermocline shoaled under the influence of El Niños, consistent with our understanding of the dynamics of the Leeuwin Current system (Feng et al., 2021).

The pre-processing of the input data via interpolation or extrapolation has dual advantages: (1) enhancing the accuracy of referent vectors in the SOM by increasing the number of good data profiles and (2) reducing the potential for errors near the bottom depth. For example, without extrapolating the temperature data to the bottom, there are blocks of anomalous warm biases near the bottom depth in the SOM-derived data (Fig. S7).

Figure 6 compares the consistency between observed and gap-filled temperature time series at three specific depths. The filled temperatures (shown in red lines) exhibit a tem-

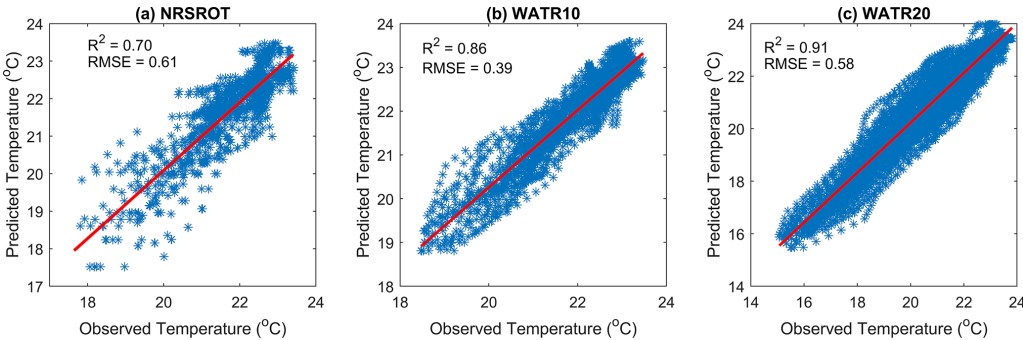

**Figure 3.** Scatterplots of observed and SOM-derived temperatures at the three moorings between 1 January and 30 May 2020, a period of 150 d. The red lines are the linear fits of the scatterplots.

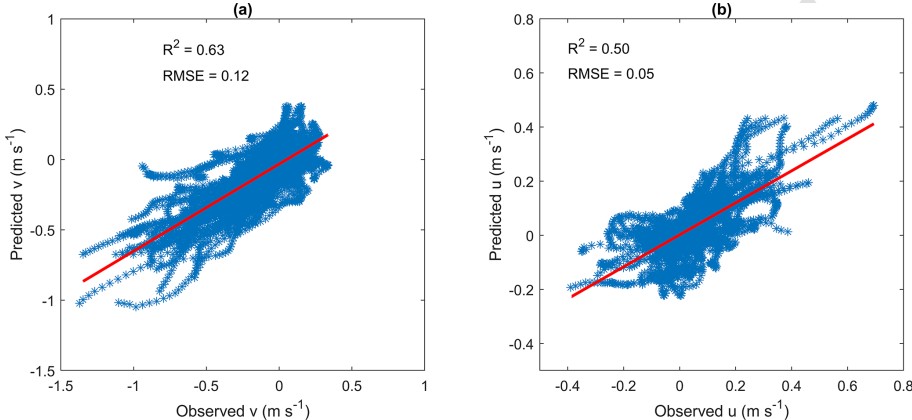

**Figure 4.** Scatterplots of observed and SOM-derived **(a)** meridional and **(b)** zonal velocities at WATR50 between May and August 2020, a period of 90 d and with a depth range of 70–450 m. The red lines are the linear fits.

perature variance similar to those of the observed time series. For example, at a depth of 95 m at WATR10 towards the end of 2011, the filled temperature is anomalously warm, reflecting the strengthened Leeuwin Current system during a La Niña period (Feng et al., 2013), as shown by the red line rising above the black dashed line. In another example, at a depth of 190 m at WATR20 during the beginning of 2010 and in the winter of 2016, the filled temperature was cooler than normal (indicated by the red line below the black dashed line) due to the shoaling of the thermocline towards the surface during El Niño episodes.

Continuous temperature time series are crucial for detecting sub-surface MHWs or MCSs that significantly affect marine ecosystems (Smale et al., 2019). Figure 7 shows the mean intensity of detected MHW or MCS events at WATR20 based on daily gap-filled temperatures. The definition of each MHW or MCS event is based on Hobday et al. (2016). An MHW (MCS) event is classified as a thermal event when its temperature exceeds the 90th percentile threshold (or falls below the 10th percentile threshold) for at least 5 d. Additionally, two consecutive events occurring within a temporal gap of less than 2 d are considered a single combined event.

This plot was created using MATLAB code for MHW and MCS detections (Zhao and Marin, 2019). Following the intense MHWs during 2011–2013 (Fig. 7a), MCSs occurred from 2016 to 2020, contributing to the recovery of impacted marine ecosystems (Fig. 7b). Many of the events are sub-surface or bottom-intensified, which are less detectable from the ocean surface based on satellite data alone.

To highlight the role of data products in detecting sub-surface MHWs, we examine several representative cases at three specific depths of different moorings: NRSROT-40m, WATR10-80m, and WATR20-100m (Fig. 8). We also analyse the meridional component of velocity at these depths to explore the roles of ocean currents in contributing to MHWs. In this study, different categories of MHWs are defined based on multiples of the local difference between the climatological mean and the 90th percentile (Hobday et al., 2018). The magnitude scale descriptors classify MHWs as moderate (between one and two multiples, Category I), strong (two to three multiples, Category II), severe (three to four multiples, Category III), and extreme (more than four multiples, Category IV). A MHW at 40 m depth at NRSROT lasted for 9 d in September 2020, with a maximum intensity of

1.5 °C, and was classified as a moderate (Category I) MHW
(Fig. 8a). During this period, the current was directed south-
ward (Fig. 8b). A MHW at 100 m depth at WATR10 lasted
for a relatively longer duration of 20 d in September 2014,
with a maximum intensity of 1.9 °C, and was classified as
a strong (Category II) MHW. Although the peak current oc-
curred during the MHW event, it led to the peak temperature
anomaly within 9 d (Fig. 8d). A MHW event at 100 m depth
at WATR20 began on 13 August 2022 and lasted for 10 d
with a maximum intensity of 1.4 °C. Unlike the other events,
the peak current led to the MHW time frame, specifically
on 10 August 2022. These observations suggest that strong
southward currents often coincide with or precede MHWs by
several days. Further research is needed to clarify the impact
of the Leeuwin Current in driving sub-surface MHWs on the
Rottnest Shelf. In addition, we zoomed in on the SOM-filled
temperatures from January to July 2011, when there was a 2-
month gap at the WATR10 mooring (Fig. S8). The gap-filled
temperatures at WATR10 (Fig. S8d) enabled us to detect the
MHW events across the water column.

Overall, the gap-filled velocity data are consistent with
temporal periods of data gaps at the mooring location and the
adjacent mooring sites (Figs. S9–S11). The observed mean
vertical profiles agree well with those derived from the filled
data (Fig. S11), indicating that the SOM method accurately
reconstructed the intricate vertical structure of the LC sys-
tem.

The LC flows along the shelf break, making velocities
measured at WATR20 and WACA20 suitable for character-
izing its primary features. From the $v$-component data, the
maximum mean currents recorded at WATR20 and WACA20
are $-0.25$ and $-0.12$ m s$^{-1}$, respectively (Fig. S11d, b). Fur-
thermore, the depths corresponding to these maximum val-
ues at the two stations are 80 and 100 m, respectively. It
can be inferred that the LC decelerates and deepens as it
flows from WATR20 to WACA20. The irregular topography
around the head of Perth Canyon may contribute to this dis-
turbance (Fig. 1).

## 4   Data availability

The outcome of this research yields the in-filled data prod-
uct, which is available at https://doi.org/10.25919/myac-
yx60 (Bui and Feng, 2024). The product comprises contin-
uous daily 5 m resolution temperature and current variables
(Table 3). All of the data products are available as NetCDF
files. In addition to the main parameters such as tempera-
ture and current, we provide quality control flags that indicate
the original data sources. Specifically, we use seven flags for
SOM-filled temperatures and four flags for SOM-filled cur-
rents, as detailed in Table 3.

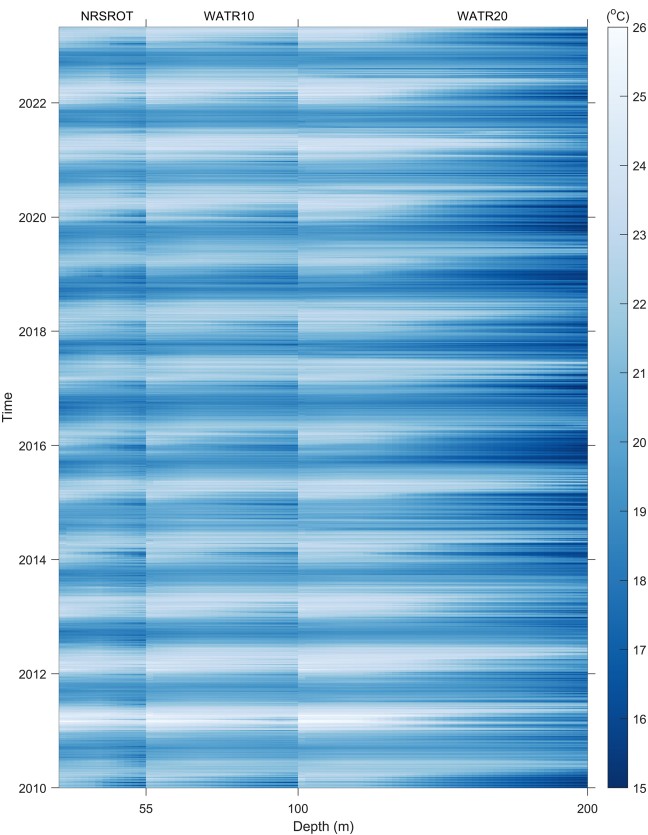

**Figure 5.** Data matrix of daily gridded, 5 m resolution gap-filled
temperatures for NRSROT, WATR10, and WATR20. The $x$ axis
shows the depth ranges of each mooring, while the $y$ axis presents
the time period from January 2010 to May 2023. Note that 0 m
follows directly after the preceding mooring. The SST data are
derived from the Regional Australian Multi-Sensor SST Analysis
(RAMSSA) version 1.0.

We provide direct links to all of the datasets used in this
study:

– unfilled gridded data – https://doi.org/10.25919/9gb1-
ne81 (Bui et al., 2023);

– satellite sea surface temperature from RAMSSA – https:
//portal.aodn.org.au (Beggs et al., 2011); and

– the Fremantle sea level from the Univer-
sity of Hawaii Sea Level Center – https:
//uhslc.soest.hawaii.edu (last access: 9 April 2025)
(https://doi.org/10.7289/V5V40S7W TS2, Caldwell et
al., 2015).

## 5   Code availability

We provide scripts in MATLAB to download and plot
the data products. These scripts are available online
(https://doi.org/10.25919/myac-yx60, Bui and Feng, 2024)

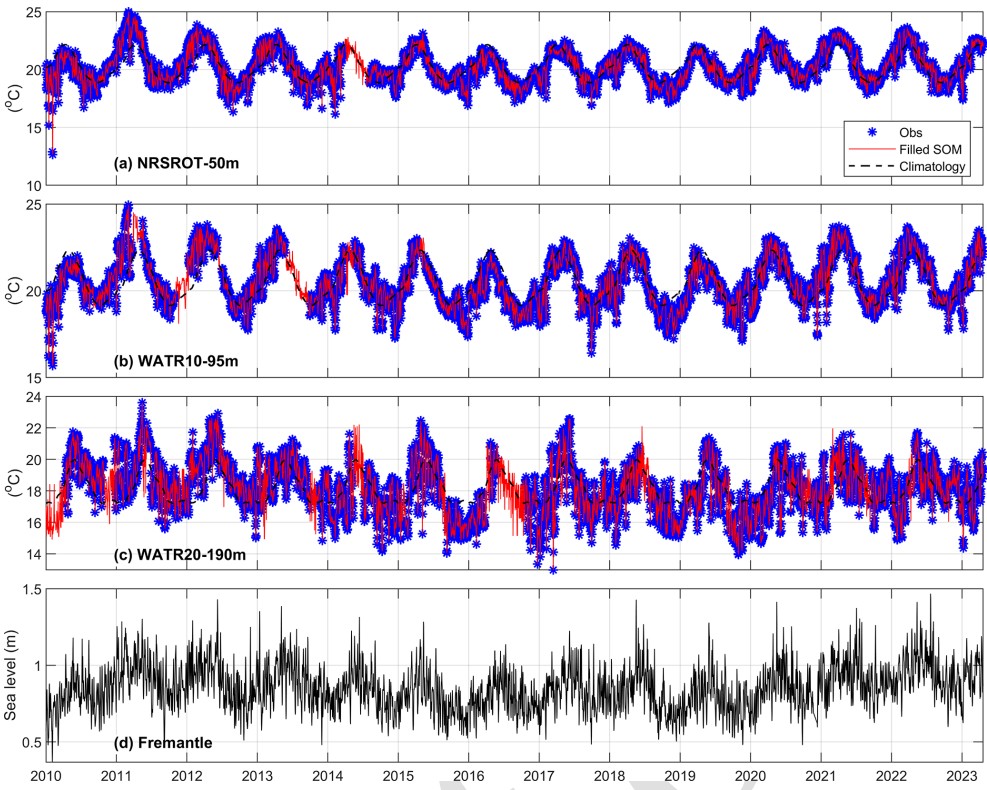

**Figure 6.** Comparison of observed and gap-filled temperature time series for **(a)** NRSROT at 50 m, **(b)** WATR10 at 95 m, and **(c)** WATR20 at 190 m. The black dashed lines show daily climatological time series at the corresponding depths. The climatological values are estimated from gap-filled data. Panel **(d)** shows the Fremantle sea level time series.

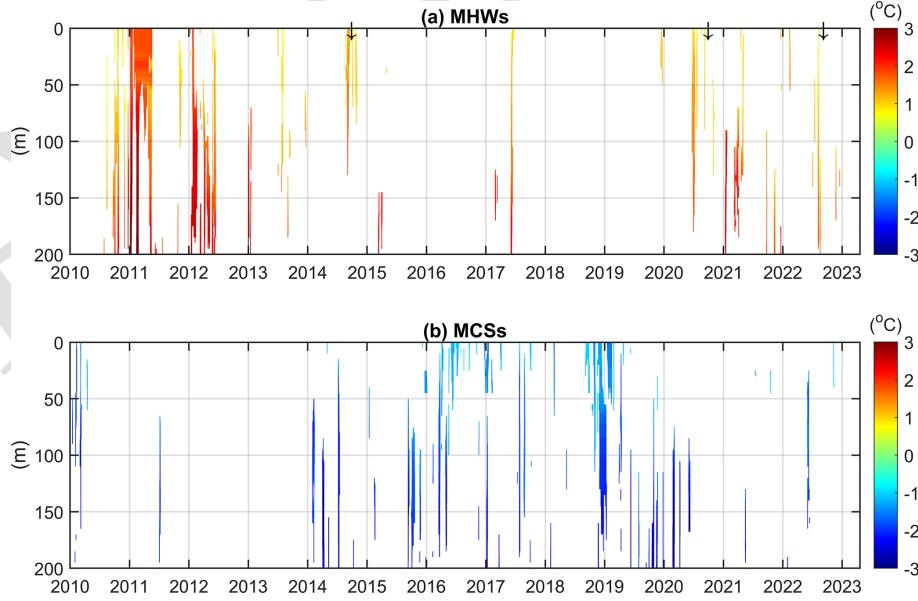

**Figure 7.** Mean intensity for the individual **(a)** MHW and **(b)** MCS events at WATR20. Estimation is based on the daily gap-filled temperature. The definition of each event follows Hobday et al. (2016). This plot is created using MATLAB code (Zhao and Marin, 2019). The threshold temperature identifying a MHW or MCS is set at the 90th and 10th percentiles, respectively. The three arrows in panel **(a)** denote the times of the MHW events shown in Fig. 8.

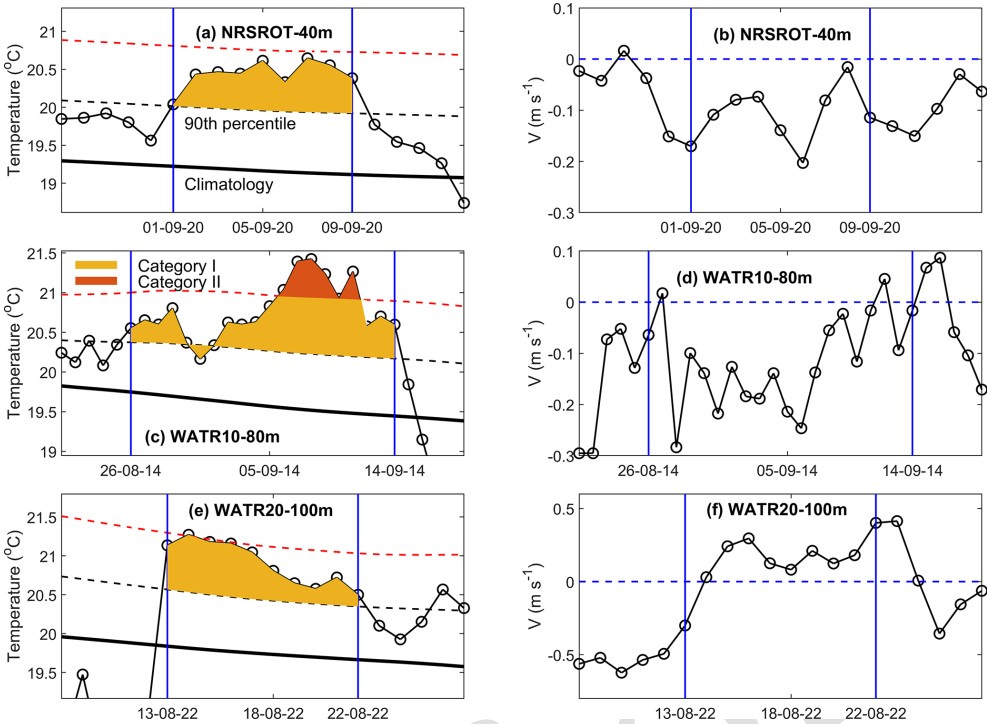

**Figure 8. (a, c, e)** Examples of marine heat waves at NRSROT-40m **(a)**, WATR10-80m **(c)**, and WATR20-100m **(e)**. The categories are moderate (yellow – category I) and strong (red – category II), as defined by Hobday et al. (2018). In panels **(a)**, **(c)**, and **(e)**, the dashed red lines are estimated as twice the 90th percentile difference from the mean climatology value. **(b, d, f)** Meridional component of the current velocity at the same time and depth as MHWs shown in the corresponding left panels. In all the panels, vertical blue lines indicate the time frame of each MHW event.

and are available under a Creative Commons Attribution 4.0 International license (CC BY 4.0).

## 6   Summary and discussion

In this research, we have employed a SOM-based method to fill significant temperature and velocity measurement gaps from a mooring array on the Rottnest Shelf off south-western Western Australia that monitors the Leeuwin Current and the associated shelf processes. We use daily temperature records from three moorings of approximately 13.5 years as well as nearly 13 years of daily current velocity records from five moorings, in conjunction with daily SST and coastal sea level at Fremantle, to train a SOM. Because this is a relatively small mooring array, we pre-process observational data using interpolation and extrapolation to have enough non-missing daily data profiles to train a SOM. Evaluated by withholding data, the RMSEs for temperature estimations at the three moorings are 0.61 °C at NRSROT, 0.39 °C at WATR10, and 0.58 °C at WATR20, respectively. The RMSEs for the meridional (along-shore) and zonal (cross-shore) velocities are 0.1 and 0.05 m s$^{-1}$. In addition, the data pre-processing brings better consistency between the observed and gap-filled data.

Since the strength of the LC is also influenced by local winds, we have evaluated the impact of including local winds during SOM training. Figure S12 presents the observed and reconstructed temperatures at the three moorings between 1 January and 30 May 2020, with local winds incorporated into the SOM training process. Compared to the case where local winds were excluded (Fig. 3), we found that including local winds resulted in a lower RMSE at NRSROT but a higher RMSE at WATR20. Overall, the differences were minimal. Local winds are important for the seasonal climatology of the Leeuwin Current. However, on interannual and intraseasonal timescales, the Leeuwin Current is more influenced by remotely forced coastal Kelvin waves, as reflected in coastal sea level variations (Feng et al., 2003; Marshall and Hendon, 2014). The effects of local winds may also have been integrated into the sea level variations.

SOM is an unsupervised learning method capable of capturing non-linear processes in the training data. Liu and Weisberg (2005) showed that the SOM method, unlike linear empirical orthogonal functions (EOFs), was able to reveal asymmetric features in the Florida Current system, such as variations in current strength and coastal jet location. However, as a statistical approach, it relies on enough realizations in the training dataset to properly capture the non-linearity. In the Rottnest Shelf region, several factors contribute to the non-linear variability in both temperature and velocity fields. Mesoscale eddies can stem from the instability of the

**Table 3.** Variables included in the in-filled data product.

| Parameter | Variable name | Unit | Description |
|---|---|---|---|
| Time | TIME | d | An array containing time information (days since 1950/01/01, 00:00:00 UTC) |
| Depth | DEPTH | m | An array containing depth levels |
| Longitude | LONGITUDE | °E | |
| Latitude | LATITUDE | °N | |
| Temperature | TEMP | °C | A matrix containing temperatures over the entire record for the whole water column |
| Temperature_quality control | TEMP_quality_control | | A matrix containing flag values that indicate the original temperature data 1. Observed temperature 2. SST 3. Interpolated temperature near the surface 4. Extrapolated temperature near the bottom 5. SOM-filled temperature near the surface 6. SOM-filled temperature in the sensor range 7. SOM-filled temperature near the bottom |
| $U$ velocity | UCUR | $\mathrm{m\,s^{-1}}$ (true east) | A matrix containing current data over the entire record for the whole water column |
| $V$ velocity | VCUR | $\mathrm{m\,s^{-1}}$ (true north) | |
| Current_quality_control | UCUR_quality_control VCUR_quality_control | | A matrix containing flag values that indicate the original current data 1. Observed current 2. Extrapolated current near the surface 3. Extrapolated current near the bottom 4. SOM-filled current |

Leeuwin Current. Intense land and sea breezes during summer amplify near-inertial currents (Mihanović et al., 2016). Additionally, the strong shear zone between the Capes Current and the Leeuwin Current in summer, as well as interactions between the strengthening of the LC and the Perth Canyon in winter, can generate sub-mesoscale eddies (Cosoli et al., 2020). SOM may well capture the mesoscale processes in the LC. Due to their randomness, however, sub-mesoscale processes may not be fully captured in daily velocities. This is reflected in the lower $R^2$ values for velocities compared to temperatures (Figs. 3 and 4).

There are weak biases in the SOM-derived product, such as a warm bias at WATR20 during the validation period from 1 January to 30 May 2020 (Fig. S6c). It is noted that this period experienced multiple marine cold spells (Fig. 7b). This systematic bias is likely due to the nature of the SOM algorithm, which tends to underestimate the magnitudes of extreme events while effectively capturing broader patterns. Future work could explore bias correction techniques to enhance accuracy.

Our continuous daily data products reveal that numerous MHW and MCS events occur below the surface, which are undetectable while using altimetry data (Fig. 7). We also find that intense MHW events are frequently related to strong southward currents at the same depth (Fig. 8). However, the role of advection temperature due to the shelf or slope LC or warm-core eddies remains unclear. Future mooring observations are needed to better understand the characteristics of MHWs and MCSs as well as the factors driving extreme temperatures.

Addressing small gaps in the mooring data appears to be a crucial step before training SOM. We have tried two other options: assigning missing values as zeros or replacing them with climatological values derived from the original data. We have experimented with these two options with an iterative approach (e.g. Sloyan et al., 2023) but found that the filled temperature time series exhibits some inconsistency, such as a block of constant values or temperature inversions. Our option of pre-processing the observational data by filling small gaps increases the number of good profiles for training. For example, 75 % of the temperature profiles are gap-free. The

method can easily be applied to fill data gaps in shelf mooring arrays with small gaps in the vertical so that little errors are introduced from linear extrapolation. For complex mooring systems with enough redundancy, the iterative completion self-organizing map (ITCOMPSOM) method outlined in Sloyan et al. (2023) could be more useful.

We have provided examples that highlight the advantages of using filled mooring data for end-users. The continuous daily temperature time series are essential for characterizing sub-surface marine heat waves and cold spells on the Rottnest Shelf, which can last from days to weeks. Furthermore, the gap-filled velocity time series from the mooring array allows researchers to capture episodic cross-shore and along-shore processes on the Rottnest Shelf, offering valuable insights into the dynamics of the Leeuwin Current and Capes Current. These mooring data products, when combined with other observational platforms such as the IMOS glider program and surface radar observations, can be integrated into ocean climate models to improve the accuracy of marine predictions for Western Australia.

**Supplement.** The supplement related to this article is available online at [the link will be implemented upon publication].

**Author contributions.** MF conceptualized and designed the study. TB and MF conducted the study, with SOM source codes provided by CCC. TB processed the data and produced the figures and first draft of the manuscript, together with the associated data products. MF and CCC reviewed and edited the manuscript.

**Competing interests.** The contact author has declared that none of the authors has any competing interests.

**Disclaimer.** Publisher's note: Copernicus Publications remains neutral with regard to jurisdictional claims made in the text, published maps, institutional affiliations, or any other geographical representation in this paper. While Copernicus Publications makes every effort to include appropriate place names, the final responsibility lies with the authors.

**Acknowledgements.** CSIRO collected the mooring data using the IMOS program. IMOS is enabled by the National Collaborative Research Infrastructure Strategy (NCRIS). The satellite SST was sourced from RAMSSA version 1.0. The daily Fremantle sea level was downloaded from the University of Hawaii's Sea Level Center. We thank Ryan Crossing, Ian Darby, Mark Snell, and Beau De Groot for the deployment and recovery of the moorings as well as Mark Snell and Miaoju Chen for the quality control of the mooring data. We thank Bernadette Sloyan for the constructive discussions and for sharing the code employed by Sloyan et al. (2023). We appreciate the feedback from the three reviewers, which helped enhance the quality of the paper.

**Review statement.** This paper was edited by Salvatore Marullo and reviewed by Alejandro Orfila, Giuseppe M. R. Manzella, and one anonymous referee.

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

## Remarks from the typesetter