# Peer review of "Gap-filled subsurface mooring dataset off"

_Earth System Science Data, 2024_

## Author Comment (AC1)

**Author Response:**

**Gap-filled subsurface mooring dataset off Western Australia during 2010–2023**

Toan Bui[1], Ming Feng[1], Christopher Chapman[2]

[1]CSIRO Environment, Indian Ocean Marine Research Centre, Crawley, WA, Australia

**[2]**CSIRO Environment, Hobart, Tasmania, Australia

*Correspondence to:* ming.feng@csiro.au

We would like to thank the three reviewers for their constructive comments. Below, we provide our detailed point-by-point responses to the review comments. The reviewers' comments are in black, our response is in regular blue colour, and our planned revisions in the manuscript are in *bold orange italics*.

**Reviewer #1:** Alejandro Orfila

Although the use of the Self Organized Maps for gap filling of time series is not new in ocean studies, this work provides the access of a complete dataset of velocity at temperature data at different depths in the Rottnest Shelf area. These data are of great interest, as the authors state, for the analysis of the seasonal and interannual variability of the Leeuwin Current (LC) and to assess the influence of large scale modes of variability on it. The methodology is well sound and the paper is well written although I would appreciate that some aspects should be treated in more detail.

We would like to thank the  reviewer's positive comments.

Q1: The first question that at least can be discussed is why the authors don't use local winds (at least at one station as ancillary data to train the SOM since, as they stated in the introduction, the strength of the LC is largely influenced by winds.  Also I suggest to show in Figure 6 the time series of sea level used in the study.

A1: We have added time series of the Fremantle sea level in Figure 6 (Fig. R1). The Fremantle sea level has been used as a proxy for the strength of the Leeuwin Current. Local winds are important for the seasonal climatology of the Leeuwin Current, however, on the interannual and intraseasonal time scales, the Leeuwin Current is more influenced by remotely forced coastal Kelvin waves, reflected in the coastal sea levels (Feng et al. 2003; Marshall and Hendon 2014). Following the reviewer's suggestion, we have tested using wind time series in the SOM training, and the results are mixed as shown below.

We have added Figure S9 in the Supplement document (Fig. R2) showing observed and reconstructed temperatures at the three moorings between 1/1/2020 and 30/5/2020, with the local wind included during the SOM training. Compared to the scenario where local wind was excluded (Figure 3), we found that including the local wind resulted in a lower RMSE at NRSROT but a higher RMSE at

[Figure]

**Figure R1. Comparison of observed and gap-filled temperature timeseries for a) NRSROT at 50m, b) WATR10 at 95m and c) WATR20 at 190m. The black dashed lines show daily climatological timeseries at corresponding depths. The climatological values are estimated from gap-filled data. The bottom panel shows Fremantle sea level timeseries.**

**Figure R2. Temperatures at the three moorings were observed and estimated between 1/1/2020 and 30/5/2020, with the local wind included during the SOM training. Compared to the scenario where local wind was excluded (Figure 3), we found that including the local wind resulted in a lower RMSE at NRSROT but a higher RMSE at WATR20. Overall, the differences were minimal. Therefore, we conclude that incorporating the local wind during SOM training does not significantly impact the results. The wind data was sourced from ERA5, covering the regional area of 32.5°S-31.5°S and 115°E-115.5°E.**

WATR20. Overall, the differences were minimal. Therefore, we conclude that incorporating the local wind during SOM training does not significantly affect the results.

References:

Marshall, A. G. and Hendon, H. H.: Impacts of the MJO in the Indian Ocean and on the Western Australian coast, Climate Dynamics, 42, 579-595, 10.1007/s00382-012-1643-2, 2014.

Feng, M., Meyers, G., Pearce, A., and Wijffels, S.: Annual and interannual variations of the Leeuwin Current at 32°S, J. Geophys. Res, 108, https://doi.org/10.1029/2002JC001763, 2003.

Q2: Lines 118-120. Could you please discuss why selecting these large numbers of neurons?. Are the results expected to be the same, reducing lets say 50 or 70%?.

A2: The selection of large numbers of neurons follows previous studies (e.g. Sloyan et al. 2023). We have tested different numbers of neurons. We tested three configurations for the temperature data: 500, 750, and 1000 units. The RMSE values at WATR20 were 0.60, 0.59, and 0.58 degrees, respectively. The results suggest that choosing 1000 neurons for the temperature data will not lead to overfit of the data. For the velocity data, we evaluated scenarios with 300, 400, and 500 units. The RMSE values for meridional velocity were 0.13, 0.13, and 0.12 m/s, respectively. Similarly, we have used 500 units for the velocity data. Using a lower number of neurons does not significantly change the results.

References:

Sloyan, B. M., Chapman, C. C., Cowley, R., and Charantonis, A. A.: Application of Machine Learning Techniques to Ocean Mooring Time Series Data, Journal of Atmospheric and Oceanic Technology, 40, 241-260, https://doi.org/10.1175/Jtech-D-21-0183.1, 2023.

We plan to add the following sentences in section 2.2 to address this comment.

*" According to the literature, a small number of SOM units captures the general dynamics of the system (Liu and Weisberg, 2011), while a larger number provides more detailed information (Sloyan et al., 2023). In our case, where we aimed to capture detailed information from each SOM unit and the training data contained a large number of rows, we opted for a larger number of units. For the temperature data, we tested three configurations: 500, 750, and 1000 units. However, the final results remained consistent across these tests, leading us to select 1000 units. Similarly, for the velocity data, we evaluated scenarios with 300, 400, and 500 units, ultimately choosing 500 units for velocity due to similar consistency in the results."*

Q3: Lines 182-184 ->highly speculative.

A3: We plan to modify this sentence as:

*"Further research is needed to clarify the impact of the Leeuwin Current in driving subsurface MHWs on the Rottnest Shelf."*

Q4: Besides, what can be concluded from Figure 7?

A4: This figure is to demonstrate a potential usage of the gap-filled data. It shows that the gap-filled data can be used to detect discrete events like marine heatwaves and cold spells lasting from several days to weeks. From Figure 7, we concluded that many MHW and MCS events occur sub-surface, which are undetectable while using satellite data. This conclusion is mentioned in Section 6 Summary and discussion.

**Reviewer #2:** Anonymous Referee

Q1: Are these data otherwise not available, and not assimilated in models (see e.g. Siripatana et al 2020, https://agupubs.onlinelibrary.wiley.com/doi/full/10.1029/2020JC016580 or Santana et al 2023 (Siripatana et al., 2020)/ for a different region)?

A1: We want to thank the reviewer's constructive comments. We will acknowledge the studies by Siripatana et al. (2020) and Santana et al. (2023), which examine observational data in the East Australia Current and the East Auckland Current, respectively. The data used in this study are for the Western Australian coast and are not the same as in the above cited studies. The mooring data have not been assimilated into ocean models as well as in the other two studies. We have checked with the Australian Bluelink team that these mooring data have not been assimilated in their global model.

References:
Santana, R., Macdonald, H., O'Callaghan, J., Powell, B., Wakes, S., and H. Suanda, S.: Data assimilation sensitivity experiments in the East Auckland Current system using 4D-Var, Geosci. Model Dev., 16, 3675-3698, 10.5194/gmd-16-3675-2023, 2023.
Siripatana, A., Kerry, C., Roughan, M., Souza, J. M. A. C., and Keating, S.: Assessing the Impact of Nontraditional Ocean Observations for Prediction of the East Australian Current, Journal of Geophysical Research: Oceans, 125, e2020JC016580, https://doi.org/10.1029/2020JC016580, 2020.

Q2: How large is the gain from this dataset, compared to the cited Sloyan et al (2024)?

A2: First, Sloyan et al. (2023, 2024) applied ITCOMSOM to fill gaps in the mooring array off the east coast of Australia, while our current study deals with mooring data off the west coast of Australia. So we are working on a different dataset in a quite different geographic region.

In our study, we initially followed the approach outlined by Sloyan et al. (2023, 2024). However, the iterative approach was not applicable to our region with a small number of moorings, such as the gap-filled temperatures showed abnormally high values near the ocean floor.

To resolve these issues, we developed a revised approach: For each vertical temperature profile, we first used interpolation to fill gaps in the near-surface temperatures and applied extrapolation for the bottom temperatures. Then, we trained the SOM using only complete data vectors, hence improving the accuracy of the SOM's topological structure. The entire dataset was processed in a single iteration. As shown in the manuscript, this method resulted in small RMSEs (Figure 3), and no visible errors were found in the temperature time series (Figure 6) and near-bottom temperatures.

References:
Sloyan, B. M., Chapman, C. C., Cowley, R., and Charantonis, A. A.: Application of Machine Learning Techniques to Ocean Mooring Time Series Data, Journal of Atmospheric and Oceanic Technology, 40, 241-260, https://doi.org/10.1175/Jtech-D-21-0183.1, 2023.
Sloyan, B. M., Cowley, R., and Chapman, C. C.: East Australian Current velocity, temperature and salinity data products, Scientific Data, 11, 10, https://doi.org/10.1038/s41597-023-02857-x, 2024.

Q3: Can you mention some example applications which would benefit from this dataset?

A3: We have presented one of the applications in the manuscript, the detection of marine heatwaves and cold spells from the mooring data, which are short-term bursts of high and low temperature events in the ocean. We plan to add in Section 6 some discussion about the applications:

*"We provide examples that highlight the advantages of using filled mooring data for end-users. By utilizing this filled data, climatological products are free from noisy data, ensuring the information is of high quality. The continuous daily temperature time series are essential for characterizing sub-surface marine heatwaves and cold spells on the Rottnest shelf. Furthermore, the gap-filled velocity time series from the mooring array allow researchers to estimate cross-shore and along-shore volume transport on the Rottnest shelf, offering valuable insights into the volume transport of the Leeuwin Current. These mooring data products, when combined with other observational platforms such as the IMOS glider program and surface radar observations, can be integrated into ocean-climate models to improve the accuracy of climate predictions for Western Australia."*

Q4: Percentages of missing data aggregated for each station are given (Table 2), but perhaps the dependence by depth should be provided too.

A4: In the supplement document, we will add Figure S10 (Fig. R3) to show the percentage of missing temperatures by depth and Figure S11 (Fig. R4) to show that of velocities.

[Figure]

**Figure R3. Percentage of missing temperature data at three moorings by depth.**

[Figure]

**Figure R4. Percentage of missing velocity data at five moorings by depth. Noting large missing data at the bottom due to changing depths of deployments with time.**

Q5: The amount of missing data seems generally low (<30%), so the added value of the SOM estimation is not evident.

A5: We disagree with this comment. Continuous time series records are required in many oceanography analyses, such as detecting MHWs/MCS following Hobday et al. 2016. Using the dataset with gaps may introduce biases in other calculations, such as seasonal climatology. Data-assimilating models are effective in filling data gaps, however, they are quite expensive. We believe data-driven approaches also offer great value.

References:
Hobday, A. J., Alexander, L. V., Perkins, S. E., Smale, D. A., Straub, S. C., Oliver, E. C., Benthuysen, J. A., Burrows, M. T., Donat, M. G., and Feng, M.: A hierarchical approach to defining marine heatwaves, Progress in Oceanography, 141, 227-238, https://doi.org/10.1016/j.pocean.2015.12.014, 2016.

Q6: Other methods used for mooring data (e.g. Cao et al (2015) https://journals.ametsoc.org/view/journals/atot/32/7/jtech-d-14-00221_1.xml should be discussed).

A6: We will add the citation and some relevant discussion in the Introduction.

References:
Cao, A.-Z., Li, B.-T., and Lv, X.-Q.: Extraction of internal tidal currents and reconstruction of full-depth tidal currents from mooring observations, Journal of Atmospheric and Oceanic Technology, 32, 1414-1424, 2015.

Q7: A plot showing SOM estimates should be provided, since Figure 6 is not very clear.

A7: We will add Figure S12 in the supplementary material (Fig. R5) to show a comparison between the observed and gap-filled temperatures at three moorings.

[Figure]

**Figure R5. Comparison between the observed and gap-filled temperatures in 2014 for NRSROT (a, b), WATR10 (c, d) and WATR20 (e, f). Left panels show the observed temperatures, while right panels display the gap-filled data.**

Q8: Different instruments are mentioned, but differences between them are not explained.

A8: We will add more information in Table 1, as below.

*Table 1. Summary of coastal mooring stations. NRSROT: National Reference Station west of the Rottnest Island. WACA: Western Australia Perth Canyon. WATR: Western Australia Two Rocks.*

| Station | Latitude; Longitude | Station depth (m) | Temperature | | | | ADCP | | | |
|---|---|---|---|---|---|---|---|---|---|---|
| | | | Instrument | Interval (min) | Mean sensor depths (m) | Data span | Instrument | Interval (min) | Bins x bin size | Data span |
| NRSROT-Temperature | 31.9900°S; 115.3850°E | 61 | SBE39[a] SBE37[b] | 5-15 | 27; 33; 43; 55 | 1/2010 - 5/2023 | | | | |
| NRSROT-ADCP | 32.0000°S; 115.4170°E; | 48 | | | | | RDI Workhorse 600 kHz[c]; Nortek Signature 500 kHz[d] | 15 | 11x4m | 8/2011 - 5/2023 |
| WACA20 | 31.9830°S; 115.2280°E | 200 | | | | | Nortek Signature 250 kHz[d]; Nortek Continental 190 kHz[d] | 15 | 41x5m | 8/2011 - 5/2023 |
| WATR10 | 31.6433°S; 115.2033°E | 100 | SBE39 SBE37 | 5-15 | 25; 30; 35; 40; 52; 70; 90 | 1/2010 - 5/2023 | Nortek Aquadopp 400 kHz[d]; | 15 | 17x5m | 8/2011 - 5/2023 |
| WATR20 | 31.7233°S; 115.0333°E | 200 | SBE39 SBE37 | 5-15 | 25; 35; 50; 68; 100; 125; 150; 175 | 1/2010 - 5/2023 | Nortek Continental 190 kHz[d]; Nortek Signature 250 kHz[d] | 15 | 25x8m | 8/2011 - 5/2023 |

| WATR50 | 31.7683°S; 114.9567°E | 500 | | | | | | RDI Long Ranger 75 kHz[c]; Nortek Signature 55 kHz[d] | 15 | 26x20m | 8/2011 - 5/2023 |
|---|---|---|---|---|---|---|---|---|---|---|---|

*a. SBE39 is a self-contained, autonomous temperature logger . (SBE: Sea-Bird Electronics).*

*b. SBE37 is a single-channel CTD (Conductivity, Temperature, Depth) sensor.*

*c. RDI ADCPs (Acoustic Doppler Current Profiles) are manufactured by Teledyne RD Instruments and comprise Long Ranger 75 kHz and Workhorse 600 kHz. (https://www.teledynemarine.com/rdi).*

*d. Nortek ADCPs are produced by Nortek group, including Signature 55 kHz, Continental 190 kHz, Signature 250 kHz, Aquadopp 400 kHz and Signature 500 kHz. (https://www.nortekgroup.com).*

Q9: In the SOM description, it is not clear to me whether the procedure is applied for each station or if they are aggregated. Have you tested various possibilities? Since stations are in a rather small area, how are measurements correlated?

A9: We have applied the SOM procedure to the aggregated dataset from all moorings. When there is data loss, often the whole mooring is lost, so it is not possible to apply the procedure to a single mooring. We tested scenarios with and without the local wind included in the ancillary data to address Q1 from Reviewer 1. The success of the SOM method relies on the correlations between the measurements.

For example, for temperature time series at a depth of 50m from three moorings, the estimated correlation ranges from 0.84 to 0.92.

Q10: More graphical examples should be provided to illustrate the method performances.

A10: We will add Figures S12, S13 and S14 in the supplementary material (Figs R5-7) to illustrate the SOM performances. Fig. R5 (above) shows a more extreme case, when the SOM method fills a large temperature data gap in the shallow mooring, utilising information from satellite SST and the deep mooring measurements. The explanation for the other two figures is below in the reply to Q11 and Q13.

Q11: Only a few scatter plots are shown, while more quantitative metrics need be used to assess the goodness of the SOM-based estimates, besides the numbers (RMSE) provided.

A11: We added Figure R6 (Fig. S13) showing the bias and standard deviations of average vertical temperature profiles between the SOM-derived data and observations, and we use climatology-dived

[Figure]

**Figure R6. Upper panels: Comparison of the temporal averages of observed (green dots), SOM-derived (red solid lines), and climatology (red dashed lines) vertical temperature profiles for a) NRSROT, b) WATR10 and c) WATR20 during the validation period between 1/1/2020 and 30/5/2020. Lower panels: mean bias of the reconstruction: black solid lines: bias of SOM estimate, that is, observed minus SOM-derived values; black dashed lines: bias of climatology estimate, that is, observed minus climatology values, during 1/1/2020 and 30/5/2020. Also the standard deviation of observed (blue dots), SOM-derived (magenta continous lines) and climatology (magenta dashed lines) temperature profiles for d) NRSROT, e) WATR10 and f) WATR20 are shown. For the climatology estimate, the temperatures observed during the validation period were withheld to estimate the daily climatology based on an 11-day moving window (Hobday et al., 2016).**

data as a reference. The mean observed (green dots), filled SOM (red continuous lines), and reconstructed climatology (red dashed lines) vertical temperature profiles at three stations during the validation period are shown in Figure R6 a-c. We calculated the residual temperatures by subtracting the filled SOM values from the observed values (Fig. R6d-f). Additionally, we assessed the performance of the SOM by evaluating the standard deviation for both the observed and reconstructed temperatures (Fig. R6d-f). In general, the mean vertical temperature profiles obtained using the SOM method have good consistencies with the observed data, compared with those from the climatology method (Fig. R6a-c). Furthermore, the standard deviations of the observed temperatures are close to those derived from SOM (Fig. R6d-f).

Q12: in section 2.3. A baseline method, e.g. climatology or AR process, should be compared.

A12: Addressed in Q11, using a climatology approach.

Q13: Perhaps analysis can be shown both for 'average' conditions and extremes, such as MHW/MCS states, and Fig S7 can serve as a starting point. However, what do you do in such plots when two profiles are overlapping? Are you showing an average?

A13: A case of extreme warm temperature conditions, or MHW is shown in Figure 8. We didn't analyse the MHW and MCS conditions exhaustively as the focus of the manuscript is on the introduction of the new dataset and allows any interested readers to explore the dataset.

[Figure]

**Figure R7. Upper panels: Comparison of the mean observed (green dots) and SOM-filled (red continous lines) vertical temperature profiles for a) NRSROT, b) WATR10 and c) WATR20 for the entire dataset. Lower panels: Black solid lines: observed values minus SOM-filled mean values; the standard deviation of observed (blue dots), SOM-filled (magenta solid lines) temperature profiles for d) NRSROT, e) WATR10 and f) WATR20.**

We add Figure S14 (Fig. R7) showing average conditions for temperatures.

Figure R7 presents a comparison of the mean observed and SOM-filled vertical temperatures at three moorings over the entire duration. The mean observed temperatures closely match the filled SOM values (Fig. R7a-c). Consequently, the residuals, calculated by subtracting the filled SOM values from the observed temperatures, are nearly zero (Fig. R7d-f). Notably, the standard deviations of both the observed and predicted temperatures at NRSROT and WATR10 are slightly higher at the surface compared to the bottom. In contrast, at WATR20, the temperature standard deviations near the surface are lower than those near the bottom (Fig. R7d-f), which is explained by the fact that many MHW/MCS events often occur near the bottom at WATR20 (Fig. 7 in the manuscript).

Q14: Detection of MHW/MCS events seems a reasonable application, but referencing and context seems missing. Please revise this part.

A14: We will update in the text (section 3 Data application) to clarify this comment:

*"The definition of each MHW or MCS event is based on Hobday et al. (2016). An MHW (MCS) event is classified as a thermal event when its temperature exceeds the 90th percentile threshold (or falls below the 10th percentile threshold) for at least 5 days. Additionally, two consecutive events occurring within a temporal gap of less than two days are considered a single combined event. This plot is performed using MATLAB code (Zhao and Marin, 2019)."*

Q15: L19 long term trends with less than 15 years of data is debatable.

A15: Corrected " long-term" to "decadal".

Q16: L27 CSIRO undefined.

[Figure]

**Figure R8. Bathymetry map and mooring locations (red circles) on the Rottnest Shelf. (a) Velocities estimated from measurements, with black arrows representing the mean state of vertically averaged velocities. The 0-200m average is used for the WATR50 mooring. The three dashed lines represent the 50m, 200m, and 500m contours. Black circles indicate the location of the Fremantle tide gauge station. Note that NRSROT consists of two separate moorings. (b) Schematic of current systems, with red arrows denoting the Leeuwin Current and blue arrows indicating the direction of the wind-driven Capes Current.**

A16: Done. Commonwealth Scientific and Industrial Research Organisation.

Q17: Fig 1 currents should be plotted from analysis data, rather than sketched manually.

A17: We will update Figure 1 (Fig. R8) showing vector currents estimated from measurements (left panel) and schematic of the current systems (right panel).

Q18: L33 You are referring to the Ningaloo here?

A18: We do not mention Ningaloo.

Q19: L45 Explain acronyms, such as SBE, in the caption. They are expanded in some cases later but the table is unclear as it stands. What's ADCP?

A19: Addressed in Q8.

Q20: Table 1 it would good to expand acronyms for locations here.

A20: Addressed in Q8.

Q21: L62 (and elsewhere) typo 'Euclidean'.

A21: Fixed.

Q22: L77 not sure what 'completion' means here.

A22: We will reword this. It means adding SST to the dataset at each mooring and then extending mooring temperature data to the sea surface by linear interpolation.

Q23: Table 2 does an empty cell mean zero?

A23: We will add in the caption of Table 2 and added crossed-out signals at empty cells:

*"Note that temperature profiles are not available at WACA20 and WATR50."*

Q24: L110 please either use lower or uppercase for MATLAB consistently.

A24: Fixed. We corrected MATLAB consistently.

Q25: Fig 2 I wonder if showing actual examples could be more informative. For example, a case with a small fraction of missing data and a more difficult one (of course from the validation set, so to compare with ground truth).

A25: We addressed this comment by adding Figures S12-14 (Figs. R5-7).

Q26: Fig 3a looks quite worse than the other two. Please explain why.

A26: The RMSE at NRSROT, which is located on the mid shelf, is higher than those of two outer-shelf sites due to influences from local variability.

Q27: L154 please make this quantitative.

A27: We addressed this comment in Q11.

Q28: Fig 5 using white for missing data is an unfortunate choice given the colorbar. Please change this to avoid ambiguity, as in S1. Also right now it looks measurements are continuous in the vertical, which is not the case. Please use a different plot, e.g. as scatter plot.

A28: We will remove the last sentence in the caption to avoid ambiguity. We confirm that this figure displays a gap-filled temperature dataset, not a gappy dataset. We retain this plot to demonstrate the consistency of the data products over time and across different locations.

Q29: L164 If you mention La Niña, then the time series should be included. As for the comment above, not sure if the Pacific or Ningaloo. How can one anticipate this situation? Do you have a reference?

A29: We will add a citation to Feng et al. (2013) reference for this event. It is a highly anticipated scenario.

Feng, M., McPhaden, M. J., Xie, S.-P., and Hafner, J.: La Niña forces unprecedented Leeuwin Current warming in 2011, Scientific Reports, 3, 1277, https://doi.org/10.1038/srep01277, 2013.

Q30: L168 Please provide background. How do you calculate this? Please reference properly.

A30: Addressed in Q14. We will add the following references.

Hobday, A.J., Alexander, L.V., Perkins, S.E., Smale, D.A., Straub, S.C., Oliver, E.C., Benthuysen, J.A., Burrows, M.T., Donat, M.G., Feng, M. and Holbrook, N.J., 2016. A hierarchical approach to defining marine heatwaves. *Progress in oceanography*, *141*, pp.227-238.

Smale, D.A., Wernberg, T., Oliver, E.C., Thomsen, M., Harvey, B.P., Straub, S.C., Burrows, M.T., Alexander, L.V., Benthuysen, J.A., Donat, M.G. and Feng, M., 2019. Marine heatwaves threaten global biodiversity and the provision of ecosystem services. *Nature Climate Change*, *9*(4), pp.306-312.

Q31: Table 3 Time has units I guess (days). There is a misplaced bracket.

A31: Fixed.

Q32: L218 What's the empty bullet?

A32: Fixed.

Q33: L235 In this part or before you should provide references to earlier works on sea temperature extremes in the area, if any.

A33: We will add some citations here, as three works on sea temperature extremes in the study area in the third paragraph of section 1 Introduction. Three references are Benthuysen et al., 2014, Feng et al., 2013 and Feng et al., 2021.

References:
Benthuysen, J., Feng, M., and Zhong, L.: Spatial patterns of warming off Western Australia during the 2011 Ningaloo Niño: Quantifying impacts of remote and local forcing, Continental Shelf Research, 91, 232-246, https://doi.org/10.1016/j.csr.2014.09.014, 2014.
Feng, M., McPhaden, M. J., Xie, S.-P., and Hafner, J.: La Niña forces unprecedented Leeuwin Current warming in 2011, Scientific Reports, 3, 1277, https://doi.org/10.1038/srep01277, 2013.
Feng, M., Caputi, N., Chandrapavan, A., Chen, M., Hart, A., and Kangas, M.: Multi-year marine cold-spells off the west coast of Australia and effects on fisheries, Journal of Marine Systems, 214, 103473, https://doi.org/10.1016/j.jmarsys.2020.103473, 2021.

Q34: L240 is it reasonable to assign 0 degrees Celsius in the water column?

A34: This is a pre-process training in SOM, not a final step to fill gaps. In the pre-process training, NaN values are assigned to zeros values so that SOM can create, initialize, and train the maps as requirements of the SOM package.

Q35: L246 I am confused, aren't you using ITCOMPSOM as stated earlier?

A35. We used the original SOM method, not ITCOMPSOM. We remove the redundant words in section 2.2 about the SOM method.

Q36: L325 what is '2015' here?

A36: Fixed.

**Reviewer #3:** Giuseppe M.R. Manzella

We want to thank the reviewer's constructive comments.

Q1: There are always essential elements to consider, but which do not seem to be well clarified in the article: how much does the non-linearity and variability of phenomena in the coastal area weigh on the method?

Figures 3 and 4 should be discussed on the basis of point 1. Before even getting to them I was in fact convinced that the method worked well with parameters such as temperature (or even salinity) but would have had significant errors with velocities.

A1: We appreciate this comment. SOM is an unsupervised learning method that is capable of capturing nonlinear processes in the training data. However, as a statistical method, it relies on enough realizations in the training dataset to properly capture the nonlinearity. Liu and Weisberg (2005) showed that the SOM method, unlike the linear EOF, was able to reveal asymmetric features in the Florida Current system, such as variations in current strength and coastal jet location. Many factors contribute to the non-linear variability in both temperatures and velocities on the Rottnest shelf. Intense land-sea breezes during the summer drive amplification of near-inertial currents (Mihanovic, 2016). Additionally, the strong shear zone between the Capes Current and the Leeuwin Current during summer as well as interactions between the strengthening of the LC and the Perth Canyon in winter, may generate sub-mesoscale eddies. Due to their randomness, these non-linear processes may not be well captured in the daily temperature and velocity training datasets. This is especially true for the current velocity. We will further clarify these points when we revise our manuscript.

 In line with our findings, Sloyan et al. (2023) observed that although the R-squared for filled temperatures is nearly 1, the values for velocities were lower (0.7 for zonal velocity and 0.8 for meridional velocity, as shown in their Figures 6 and 8).

References:
Mihanovic, H.; Pattiaratchi, C.B.; Verspecht, F. Diurnal sea breezes force near-inertial waves along Rottnest continental shelf, Southwestern Australia. J. Phys. Oceanogr. 2016, 46, 3487–3508.
Liu, Y. and Weisberg, R. H.: Patterns of ocean current variability on the West Florida Shelf using the self-organizing map, Journal of Geophysical Research: Oceans, 110, Artn C06003 10.1029/2004jc002786, 2005.
Sloyan, B. M., Chapman, C. C., Cowley, R., and Charantonis, A. A.: Application of Machine Learning Techniques to Ocean Mooring Time Series Data, Journal of Atmospheric and Oceanic Technology, 40, 241-260, https://doi.org/10.1175/Jtech-D-21-0183.1, 2023.

Q2: Are the data sufficiently representative of the physical state of the sea ? (perhaps the answers are in the articles cited by the authors, but a brief summary would have been very useful).

A2: The dataset used here is sufficient to capture the dominant alongshore and cross-shore processes on the Rottnest Shelf from intraseasonal to interannual time scales (Feng et al., 2024 and relevant studies cited in this manuscript, such as Feng et al. 2013; Benthuysen et al. 2014). For example, the mooring observations capture the Leeuwin Current variability, and the Capes Current, driven by strong southerly winds, flows northward, primarily confined to the middle shelf (20-50 m).

As for the sub-mesoscale processes on the shelf and processes associated with the Perth Canyon, they are not well captured by the mooring array. These processes would cause errors in the SOM calculations and gap-filling, resulting in high uncertainty in the mapped velocity fields. A brief overview of the dynamics in the study area is provided in Section 1. We plan to add the above point to further address this comment.

References:
Ming Feng, Toan Bui, Jessica Benthuysen, 2024. Seasonal climatology of the Leeuwin Current – Capes Current system derived from moored observations off southwest Australia. Journal of Geophysical Research: Oceans (Under review).

Q3: Line 77. Satellite data are used to extend the temperature to the surface. Since these data are part of sea truth exercises, a very brief presentation of associated precisions and uncertainties would be useful.

A3: We will add following sentences in section 2.1.1 Temperature to clarify this comment:

*"The RAMSSA system combines SST data from infrared and microwave sensors on polar-orbiting satellites with in situ measurements to generate daily foundational SST estimates. These estimates show significantly lower standard deviation compared to existing regional SST analyses. The absence of bias correction in the data input into RAMSSA has minimal impact north of 40°S where RAMSSA is on average within ±0.07 °C of other multi-sensor SST analyses. However, south of 40°S, RAMSSA is, on average, 0.09°C to 0.25°C warmer than the bias-corrected SST analyses studied (Beggs et al., 2011). These errors are much smaller than those estimated by SOM (Fig. 3).*

References:
Beggs, H., Zhong, A., Warren, G., Alves, O., Brassington, G., and Pugh, T.: RAMSSA—An operational, high-resolution, regional Australian multi-sensor sea surface temperature analysis over the Australian region, Australian Meteorological and Oceanographic Journal, 61, 1, 2011.

Q4: Line 78. The temperature in each mooring is extended to the surface with a linear interpolation. No problem with the seasonal thermocline?

A4: We looked at some CTD profiles in the study area, and we found that the temperature decreased almost linearly with depth in the near surface layer (top 30m), therefore seasonal thermocline may not pose a significant issue.

Q5: The data are interesting and should be published. But I agree with one of the other referee: possible applications of gap filled data should be discussed, not only on heat waves.

A5: This is addressed in the reply to Q3 from Reviewer#2.

---

## Author Response (AR1)

**Author Response:**

**Gap-filled subsurface mooring dataset off Western Australia during 2010–2023**

Toan Bui[1], Ming Feng[1], Christopher Chapman[2]

[1]CSIRO Environment, Indian Ocean Marine Research Centre, Crawley, WA, Australia

**[2]**CSIRO Environment, Hobart, Tasmania, Australia

*Correspondence to:* ming.feng@csiro.au

We would like to thank the three reviewers for their constructive comments. Below, we provide our detailed point-by-point responses to the review comments. The reviewers' comments are in black, our response is in regular blue colour, and our revisions in the manuscript are in ***bold orange italics***.

**Reviewer #1:** Alejandro Orfila

Although the use of the Self Organized Maps for gap filling of time series is not new in ocean studies, this work provides the access of a complete dataset of velocity at temperature data at different depths in the Rottnest Shelf area. These data are of great interest, as the authors state, for the analysis of the seasonal and interannual variability of the Leeuwin Current (LC) and to assess the influence of large scale modes of variability on it. The methodology is well sound and the paper is well written although I would appreciate that some aspects should be treated in more detail.

We would like to thank the reviewer's positive comments.

Q1: The first question that at least can be discussed is why the authors don't use local winds (at least at one station as ancillary data to train the SOM since, as they stated in the introduction, the strength of the LC is largely influenced by winds. Also I suggest to show in Figure 6 the time series of sea level used in the study.

A1: We added the time series of the Fremantle sea level in Figure 6 (Fig. R1). The Fremantle sea level has been used as a proxy for the strength of the Leeuwin Current. Local winds are important for the seasonal variations of the Leeuwin Current, however, on the interannual and intraseasonal time scales, the Leeuwin Current is more influenced by remotely forced coastal Kelvin waves, reflected in the coastal sea levels (Feng et al. 2003; Marshall and Hendon 2014). Following the reviewer's suggestion, we have tested using wind time series in the SOM training, and the results are mixed as shown below.

We added Figure S14 in the Supplement document (Fig. R2) showing observed and reconstructed temperatures at the three moorings between 1/1/2020 and 30/5/2020, with the local wind included during the SOM training. Compared to the scenario where the local wind was excluded (Figure 3), we found that including the local wind resulted in a lower RMSE at NRSROT but a higher RMSE at

[Figure]

**Figure R1. Comparison of observed and gap-filled temperature timeseries for a) NRSROT at 50m, b) WATR10 at 95m and c) WATR20 at 190m. The black dashed lines show daily climatological timeseries at corresponding depths. The climatological values are estimated from gap-filled data. The bottom panel shows Fremantle sea level timeseries.**

**Figure R2. Temperatures at the three moorings were observed and estimated between 1/1/2020 and 30/5/2020, with the local wind included during the SOM training. Compared to the scenario where local wind was excluded (Figure 3), we found that including the local wind resulted in a lower RMSE at NRSROT but a higher RMSE at WATR20. Overall, the differences were minimal. Therefore, we conclude that incorporating the local wind during SOM training does not significantly impact the results. The wind data was sourced from ERA5, covering the regional area of 32.5°S-31.5°S and 115°E-115.5°E.**

WATR20. Overall, the differences were minimal. Therefore, we conclude that incorporating the local

wind during SOM training does not significantly affect the results, as its effects may have already

been integrated into the sea level variations.

We added one paragraph in Section 6 to clarify this comment.

*"Since the strength of the LC is also influenced by local winds, we have evaluated the impact of including local winds during SOM training. Figure S14 presents the observed and reconstructed temperatures at the three moorings between January 1, 2020, and May 30, 2020, with local winds incorporated into the SOM training process. Compared to the case where local winds were excluded (Fig. 3), we found that including local winds resulted in a lower RMSE at NRSROT but a higher RMSE at WATR20. Overall, the differences were minimal. Local winds are important for the seasonal climatology of the Leeuwin Current, however, on interannual and intraseasonal time scales, the Leeuwin Current is more influenced by remotely forced coastal Kelvin waves, as reflected in coastal sea levels (Feng et al. 2003; Marshall and Hendon, 2014). The effects of local winds may also have been integrated into the sea level variations."*

References:
Marshall, A. G. and Hendon, H. H.: Impacts of the MJO in the Indian Ocean and on the Western Australian coast, Climate Dynamics, 42, 579-595, 10.1007/s00382-012-1643-2, 2014.
Feng, M., Meyers, G., Pearce, A., and Wijffels, S.: Annual and interannual variations of the Leeuwin Current at 32°S, J. Geophys. Res, 108, https://doi.org/10.1029/2002JC001763, 2003.

Q2: Lines 118-120. Could you please discuss why selecting these large numbers of neurons?. Are the results expected to be the same, reducing lets say 50 or 70%?.

A2: The selection of large numbers of neurons follows previous studies (e.g. Sloyan et al. 2023). We have tested different numbers of neurons. We tested three configurations for the temperature data: 500, 750, and 1000 units. The RMSE values at WATR20 were 0.60°C, 0.59°C, and 0.58°C, respectively. The results suggest that choosing 1000 neurons for the temperature data will not lead to overfitting the data. For the velocity data, we evaluated scenarios with 300, 400, and 500 units. The RMSE values for meridional velocity were 0.13, 0.13, and 0.12 m/s, respectively. Similarly, we have used 500 units for the velocity data. Using a lower number of neurons does not significantly change the results.

References:
Sloyan, B. M., Chapman, C. C., Cowley, R., and Charantonis, A. A.: Application of Machine Learning Techniques to Ocean Mooring Time Series Data, Journal of Atmospheric and Oceanic Technology, 40, 241-260, https://doi.org/10.1175/Jtech-D-21-0183.1, 2023.

We have modified section 2.2 to address this comment.

*"The number of units in the SOM is specified prior to the training process. According to the literature, a small number of SOM units is useful in capturing the general features of the system (Liu and Weisberg, 2011), while a larger number provides more detailed information and is more suitable for data gap filling (Sloyan et al., 2023). In our case, where we aimed to capture detailed information from the training data contained a large number of profiles, we opted for a larger number of units, 1000 units for the temperature data and 500 units for the velocity data. Using lower numbers of units only had minor effects on the results."*

Q3: Lines 182-184 ->highly speculative.

A3: We modified this sentence as:

*"Further research is needed to clarify the impact of the Leeuwin Current in driving subsurface MHWs on the Rottnest Shelf."*

Q4: Besides, what can be concluded from Figure 7?

A4: This figure is to demonstrate a potential usage of the gap-filled data. It shows that the gap-filled data can be used to detect discrete events like marine heatwaves and cold spells lasting from several days to weeks. From Figure 7, we concluded that many MHW and MCS events occur sub-surface, which are undetectable while using satellite data. This conclusion is mentioned in Section 6 Summary and discussion.

**Reviewer #2:** Anonymous Referee

Q1: Are these data otherwise not available, and not assimilated in models (see e.g. Siripatana et al 2020, https://agupubs.onlinelibrary.wiley.com/doi/full/10.1029/2020JC016580 or Santana et al 2023 / for a different region)?

A1: We want to thank the reviewer's constructive comments. We will acknowledge the studies by Siripatana et al. (2020) and Santana et al. (2023), which examine observational data in the East Australia Current and the East Auckland Current, respectively. The data used in this study are for the Western Australian coast and are not the same as in the above-cited studies. The mooring data have not been assimilated into ocean models as well as in the other two studies. We have checked with the Australian Bluelink team that these mooring data have not been assimilated into their global model.

References:
Santana, R., Macdonald, H., O'Callaghan, J., Powell, B., Wakes, S., and H. Suanda, S.: Data assimilation sensitivity experiments in the East Auckland Current system using 4D-Var, Geosci. Model Dev., 16, 3675-3698, 10.5194/gmd-16-3675-2023, 2023.
Siripatana, A., Kerry, C., Roughan, M., Souza, J. M. A. C., and Keating, S.: Assessing the Impact of Nontraditional Ocean Observations for Prediction of the East Australian Current, Journal of Geophysical Research: Oceans, 125, e2020JC016580, https://doi.org/10.1029/2020JC016580, 2020.

Q2: How large is the gain from this dataset, compared to the cited Sloyan et al (2024)?

A2: First, Sloyan et al. (2023, 2024) applied ITCOMSOM to fill gaps in the mooring array off the east coast of Australia, while our current study deals with mooring data off the west coast of Australia. So we are working on a different dataset in a quite different geographic region.

In our study, we initially followed the approach outlined by Sloyan et al. (2023, 2024). However, the iterative approach was not applicable to our region with a small number of moorings, such as the gap-filled temperatures showing abnormally high values near the ocean floor (Fig. S8, supplementary materials).

To resolve these issues, we developed a revised approach: For each vertical temperature profile, we first used interpolation to fill gaps in the near-surface temperatures and applied extrapolation for the bottom temperatures. Then, we trained the SOM using only complete data vectors, hence improving the accuracy of the SOM's topological structure. The entire dataset was processed in a single iteration. As shown in the manuscript, this method resulted in small RMSEs (Figure 3), and no visible errors were found in the near-bottom temperatures (Figure 5) and the temperature time series (Figure 6).

References:
Sloyan, B. M., Chapman, C. C., Cowley, R., and Charantonis, A. A.: Application of Machine Learning Techniques to Ocean Mooring Time Series Data, Journal of Atmospheric and Oceanic Technology, 40, 241-260, https://doi.org/10.1175/Jtech-D-21-0183.1, 2023.

Sloyan, B. M., Cowley, R., and Chapman, C. C.: East Australian Current velocity, temperature and salinity data products, Scientific Data, 11, 10, https://doi.org/10.1038/s41597-023-02857-x, 2024.

Q3: Can you mention some example applications which would benefit from this dataset?

A3: We have presented one of the applications in the manuscript, the detection of marine heatwaves and cold spells from the mooring data, which are short-term bursts of high and low temperature events in the ocean. We added in Section 6 some discussion about the applications:

*"We have provided examples that highlight the advantages of using filled mooring data for end-users. The continuous daily temperature time series are essential for characterizing sub-surface marine heatwaves and cold spells on the Rottnest shelf, which can last from days to weeks. Furthermore, the gap filled velocity time series from the mooring array allows researchers to capture episodical cross-shore and along-shore processes on the Rottnest shelf, offering valuable insights into the dynamics of the Leeuwin Current and Capes Current. These mooring data products, when combined with other observational platforms such as the IMOS glider program and surface radar observations, can be integrated into ocean-climate models to improve the accuracy of marine predictions for Western Australia."*

Q4: Percentages of missing data aggregated for each station are given (Table 2), but perhaps the dependence by depth should be provided too.

A4: In the supplement document, we added Figure S4 (Fig. R3) to show the percentage of missing temperatures by depth and Figure S5 (Fig. R4) to show that of velocities.

[Figure]

**Figure R3. Percentage of missing temperature data at three moorings by depth.**

[Figure]

**Figure R4. Percentage of missing velocity data at five moorings by depth. Noting large missing data at the bottom due to changing depths of deployments with time.**

Q5: The amount of missing data seems generally low (<30%), so the added value of the SOM estimation is not evident.

A5: We disagree with this comment. Continuous time series records are required in many oceanography analyses, such as detecting MHWs/MCS following Hobday et al. 2016. Using the dataset with gaps may introduce biases in other calculations, such as seasonal climatology. Data-assimilating models are effective in filling data gaps, however, they are quite expensive. We believe data-driven approaches also offer great value.

References:
Hobday, A. J., Alexander, L. V., Perkins, S. E., Smale, D. A., Straub, S. C., Oliver, E. C., Benthuysen, J. A., Burrows, M. T., Donat, M. G., and Feng, M.: A hierarchical approach to defining marine heatwaves, Progress in Oceanography, 141, 227-238, https://doi.org/10.1016/j.pocean.2015.12.014, 2016.

Q6: Other methods used for mooring data (e.g. Cao et al (2015) https://journals.ametsoc.org/view/journals/atot/32/7/jtech-d-14-00221_1.xml should be discussed).

A6: We added the citation and some relevant discussion in the Introduction.

References:
Cao, A.-Z., Li, B.-T., and Lv, X.-Q.: Extraction of internal tidal currents and reconstruction of full-depth tidal currents from mooring observations, Journal of Atmospheric and Oceanic Technology, 32, 1414-1424, 2015.

Q7: A plot showing SOM estimates should be provided, since Figure 6 is not very clear.

A7: We provided Fig. R5 (Figure S9) to provide a zoomed-in view of SOM-filled temperatures in 2011. For example, SOM effectively filled a large data gap at WATR10 spanned nearly two months from March to May 2011.

We added in Section 3 some discussion about this Figure.

*"In addition, we zoomed in on the SOM-filled temperatures from January to July 2011 when there was a two-month gap at the WATR10 mooring (Fig. S9). The gap-filled temperatures at WATR10 (Fig. S9d) enabled us to detect the MHW events across the water column."*

[Figure]

**Figure R5. Comparison of observed and gap-filled temperatures in 2011 for NRSROT (a, b), WATR10 (c, d), and WATR20 (e, f). The left panels depict observed temperatures, while the right panels display gap-filled data. Black crosses in panels (b), (d), and (f) mark days classified as marine heatwaves (MHW), characterized by observed temperatures above the 90th percentile for at least five consecutive days (Hobday et al., 2016). Magenta dashed lines indicate the onset of missing temperature blocks at WATR10, and magenta solid lines denote their conclusion. Despite the missing data, the gap-filled temperatures at WATR10 successfully reconstructed the MHW event (panel d), demonstrating the ability of the SOM to accurately recover extreme warm temperatures.**

Q8: Different instruments are mentioned, but differences between them are not explained.

A8: We added more information in Table 1, as below.

*Table 1. Summary of coastal mooring stations. NRSROT: National Reference Station west of the Rottnest Island. WACA: Western Australia Perth Canyon. WATR: Western Australia Two Rocks.*

| Station | Latitude; Longitude | Station depth (m) | Temperature | | | | ADCP | | | |
|---|---|---|---|---|---|---|---|---|---|---|
| | | | Instrument | Interval (min) | Mean sensor depths (m) | Data span | Instrument | Interval (min) | Bin numbers x bin size | Data span |
| NRSROT-Temperature | 31.9900°S; 115.3850°E | 61 | SBE39[a] SBE37[b] | 5-15 | 27; 33; 43; 55 | 1/2010 - 5/2023 | | | | |
| NRSROT-ADCP | 32.0000°S; 115.4170°E; | 48 | | | | | RDI Workhorse 600 kHz[c]; Nortek Signature 500 kHz[d] | 15 | 11x4m | 8/2011 - 5/2023 |
| WACA20 | 31.9830°S; 115.2280°E | 200 | | | | | Nortek Signature 250 kHz[d]; Nortek Continental 190 kHz[d] | 15 | 41x5m | 8/2011 - 5/2023 |
| WATR10 | 31.6433°S; 115.2033°E | 100 | SBE39 SBE37 | 5-15 | 25; 30; 35; 40; 52; 70; 90 | 1/2010 - 5/2023 | Nortek Aquadopp 400 kHz[d]; | 15 | 17x5m | 8/2011 - 5/2023 |
| WATR20 | 31.7233°S; 115.0333°E | 200 | SBE39 SBE37 | 5-15 | 25; 35; 50; 68; 100; 125; 150; 175 | 1/2010 - 5/2023 | Nortek Continental 190 kHz[d]; Nortek Signature 250 kHz[d] | 15 | 25x8m | 8/2011 - 5/2023 |

| WATR50 | 31.7683°S; 114.9567°E | 500 | | | | | RDI Long Ranger 75 kHz[c]; Nortek Signature 55 kHz[d] | 15 | 26x20m | 8/2011 - 5/2023 |
|---|---|---|---|---|---|---|---|---|---|---|

*a. SBE39 is a self-contained, autonomous temperature logger . (SBE: Sea-Bird Electronics).*

*b. SBE37 is a single-channel CTD (Conductivity, Temperature, Depth) sensor.*

*c. RDI ADCPs (Acoustic Doppler Current Profiles) are manufactured by Teledyne RD Instruments and comprise Long Ranger 75 kHz and Workhorse 600 kHz. (https://www.teledynemarine.com/rdi).*

*d. Nortek ADCPs are produced by Nortek group, including Signature 55 kHz, Continental 190 kHz, Signature 250 kHz, Aquadopp 400 kHz and Signature 500 kHz. (https://www.nortekgroup.com).*

Q9: In the SOM description, it is not clear to me whether the procedure is applied for each station or if they are aggregated. Have you tested various possibilities? Since stations are in a rather small area, how are measurements correlated?

A9: We have applied the SOM procedure to the aggregated dataset from all moorings. When there is data loss, often the whole mooring is lost, so it is not possible to apply the procedure to a single mooring. We tested scenarios with and without the local wind included in the ancillary data to address Q1 from Reviewer 1. The success of the SOM method relies on the correlations between the measurements.

For example, for temperature time series at a depth of 50m from three moorings, the estimated correlation ranges from 0.84 to 0.92.

Q10: More graphical examples should be provided to illustrate the method performances.

A10: We added Fig. R5 (Fig. S9), Fig. R6 (Fig. S7) and Fig. R7 to illustrate the SOM performances.

The explanation for the other two figures are below in the replies to Q11 and Q13.

Q11: Only a few scatter plots are shown, while more quantitative metrics need be used to assess the goodness of the SOM-based estimates, besides the numbers (RMSE) provided.

A11: We added Fig. R6 (Fig. S7) showing the bias and standard deviations of average vertical temperature profiles between the SOM-derived data and observations, and we use climatology-

[Figure]

**Figure R6. Upper panels: Comparison of the temporal averages of observed (green dots), SOM-derived (red solid lines), and climatology (red dashed lines) vertical temperature profiles for a) NRSROT, b) WATR10 and c) WATR20 during the validation period between 1/1/2020 and 30/5/2020. Lower panels: mean bias of the reconstruction: black solid lines: bias of SOM estimate, that is, observed minus SOM-derived values; black dashed lines: bias of climatology estimate, that is, observed minus climatology values, during 1/1/2020 and 30/5/2020. Also the standard deviation of observed (blue dots), SOM-derived (magenta continous lines) and climatology (magenta dashed lines) temperature profiles for d) NRSROT, e) WATR10 and f) WATR20 are shown. For the climatology estimate, the temperatures observed during the validation period were withheld to estimate the daily climatology based on an 11-day moving window (Hobday et al., 2016).**

derived data as a reference. The mean observed (green dots), filled SOM (red continuous lines), and reconstructed climatology (red dashed lines) vertical temperature profiles at three stations during the validation period are shown in Figure R6 a-c. We calculated the residual temperatures by subtracting the filled SOM values from the observed values (Fig. R6d-f). Additionally, we assessed the performance of the SOM by evaluating the standard deviation for both the observed and reconstructed temperatures (Fig. R6d-f). In general, the mean vertical temperature profiles obtained using the SOM method have good consistencies with the observed data, compared with those from the climatology method (Fig. R6a-c). Furthermore, the standard deviations of the observed temperatures are close to those derived from SOM (Fig. R6d-f).

We added the following sentences in section 2.3 to address this comment.

*"To assess further the accuracy of the SOM method, we compare it with the climatology method over the same validation period, as shown in Fig. S7. Overall, the mean vertical temperature profiles from the SOM method are closer to the observed data than those from the climatology method (Fig. 7a-c). As a result, the residuals from the SOM method, calculated by subtracting the filled SOM values from the observed temperatures, are smaller than the climatological residuals. Additionally, the standard deviation of the observed temperatures is closer to that of the SOM data, while it differs significantly from the climatological values (Fig. S7d-f). These findings suggest that the SOM method is more reliable than the climatology method."*

[Figure]

**Figure R7. Upper panels: Comparison of the mean observed (green dots) and SOM-filled (red continous lines) vertical temperature profiles for a) NRSROT, b) WATR10 and c) WATR20 for the entire dataset. Lower panels: Black solid lines: observed values minus SOM-filled mean values; the standard deviation of observed (blue dots), SOM-filled (magenta solid lines) temperature profiles for d) NRSROT, e) WATR10 and f) WATR20.**

Q12: in section 2.3. A baseline method, e.g. climatology or AR process, should be compared.

A12: Addressed in Q11, using a climatology approach.

Q13: Perhaps analysis can be shown both for 'average' conditions and extremes, such as MHW/MCS states, and Fig S7 can serve as a starting point. However, what do you do in such plots when two profiles are overlapping? Are you showing an average?

A13: A case of extreme warm temperature conditions, or MHW is shown in Figure 8. We didn't analyse the MHW and MCS conditions exhaustively as the focus of the manuscript is on the introduction of the new dataset and allows any interested readers to explore the dataset.

We added Fig. R7 showing average conditions for temperatures.

Figure R7 presents a comparison of the mean observed and SOM-filled vertical temperatures at three moorings over the entire duration. The mean observed temperatures closely match the filled SOM values (Fig. R7a-c). Consequently, the residuals, calculated by subtracting the filled SOM values from the observed temperatures, are nearly zero (Fig. R7d-f). Notably, the standard deviations of both the observed and predicted temperatures at NRSROT and WATR10 are slightly higher at the surface compared to the bottom. In contrast, at WATR20, the temperature standard deviations near the surface are lower than those near the bottom (Fig. R7d-f), which is explained by the fact that many MHW/MCS events often occur near the bottom at WATR20 (Fig. 7 in the manuscript).

We also added Fig. R8 (Fig. S6) showing the SOM's capability to accurately reconstruct extreme cold temperature patterns.

We added the following sentences in section 2.3 to discuss Fig. S6.

[Figure]

**Figure R8. (a) Observed and (b) SOM-predicted temperatures at WATR20 during the validation period (January 1 to May 30, 2020). Black crosses in both panels indicate days identified as marine cold spells (MCS), defined as temperatures below the 10th percentile for a minimum of five consecutive days (Hobday et al., 2016). The SOM-predicted temperatures (b) effectively captured MCS events, showcasing the SOM's capability to accurately reconstruct extreme temperature patterns.**

*"Furthermore, we evaluate the ability of the SOM to reconstruct extreme temperature patterns. As shown in Figure S6, a comparison of the observed and SOM-derived temperatures at WATR20 during the validation period (January 1 to May 30, 2020) highlights this capability. Black crosses in both panels denote days identified as marine cold spells (MCS), defined as periods where temperatures fall below the 10th percentile for at least five consecutive days (Hobday et al., 2016). SOM-derived temperatures successfully captured three bottom-intensified MCS events as in observations, demonstrating the method's reliability in reconstructing extreme cold temperature patterns."*

Q14: Detection of MHW/MCS events seems a reasonable application, but referencing and context seems missing. Please revise this part.

A14: We added in the text (section 3 Data application) to clarify this comment:

*"The definition of each MHW or MCS event is based on Hobday et al. (2016). An MHW (MCS) event is classified as a thermal event when its temperature exceeds the 90th percentile threshold (or falls below the 10th percentile threshold) for at least 5 days. Additionally, two consecutive events*

*occurring within a temporal gap of less than two days are considered a single combined event. This plot is performed using MATLAB code (Zhao and Marin, 2019)."*

Q15: L19 long term trends with less than 15 years of data is debatable.

A15: Corrected " long-term" to "decadal".

Q16: L27 CSIRO undefined.

A16: Done. Commonwealth Scientific and Industrial Research Organisation.

Q17: Fig 1 currents should be plotted from analysis data, rather than sketched manually.

A17: We updated Figure 1 (Fig. R9) showing vector currents estimated from measurements (left panel) and schematic of the current systems (right panel).

[Figure]

Figure R9. Bathymetry map and mooring locations (red circles) on the Rottnest Shelf. (a) Velocities estimated from measurements, with black arrows representing the mean state of vertically averaged velocities. The 0-200m average is used for the WATR50 mooring. The three dashed lines represent the 50m, 200m, and 500m contours. Black circles indicate the location of the Fremantle tide gauge station. Note that NRSROT consists of two separate moorings. (b) Schematic of current systems, with red arrows denoting the Leeuwin Current and blue arrows indicating the direction of the wind-driven Capes Current.

Q18: L33 You are referring to the Ningaloo here?

A18: We do not mention Ningaloo.

Q19: L45 Explain acronyms, such as SBE, in the caption. They are expanded in some cases later but the table is unclear as it stands. What's ADCP?

A19: Addressed in Q8.

Q20: Table 1 it would good to expand acronyms for locations here.

A20: Addressed in Q8.

Q21: L62 (and elsewhere) typo 'Euclidean'.

A21: Fixed.

Q22: L77 not sure what 'completion' means here.

A22: We reworded this. It means adding SST to the dataset at each mooring and then extending mooring temperature data to the sea surface by linear interpolation.

Q23: Table 2 does an empty cell mean zero?

A23: We added in the caption of Table 2 and added crossed-out signals at empty cells:

*"Note that temperature profiles are not available at WACA20 and WATR50."*

Q24: L110 please either use lower or uppercase for MATLAB consistently.

A24: Fixed. We corrected MATLAB consistently.

Q25: Fig 2 I wonder if showing actual examples could be more informative. For example, a case with a small fraction of missing data and a more difficult one (of course from the validation set, so to compare with ground truth).

A25: We addressed this comment by adding Figs. R5-7.

Q26: Fig 3a looks quite worse than the other two. Please explain why.

A26: The RMSE at NRSROT, which is located on the mid shelf, is higher than those of two outer-shelf sites due to influences from local variability.

Q27: L154 please make this quantitative.

A27: We addressed this comment in Q11.

Q28: Fig 5 using white for missing data is an unfortunate choice given the colorbar. Please change this to avoid ambiguity, as in S1. Also right now it looks measurements are continuous in the vertical, which is not the case. Please use a different plot, e.g. as scatter plot.

A28: We removed the last sentence in the caption to avoid ambiguity. We confirm that this figure displays a gap-filled temperature dataset, not a gappy dataset. We retain this plot to demonstrate the consistency of the data products over time and across different locations.

Q29: L164 If you mention La Niña, then the time series should be included. As for the comment above, not sure if the Pacific or Ningaloo. How can one anticipate this situation? Do you have a reference?

A29: We added a citation to Feng et al. (2013) reference for this event. It is a highly anticipated scenario.

Feng, M., McPhaden, M. J., Xie, S.-P., and Hafner, J.: La Niña forces unprecedented Leeuwin Current warming in 2011, Scientific Reports, 3, 1277, https://doi.org/10.1038/srep01277, 2013.

Q30: L168 Please provide background. How do you calculate this? Please reference properly.

A30: Addressed in Q14. We added the following references.

Hobday, A.J., Alexander, L.V., Perkins, S.E., Smale, D.A., Straub, S.C., Oliver, E.C., Benthuysen, J.A., Burrows, M.T., Donat, M.G., Feng, M. and Holbrook, N.J., 2016. A hierarchical approach to defining marine heatwaves. *Progress in oceanography*, *141*, pp.227-238.

Smale, D.A., Wernberg, T., Oliver, E.C., Thomsen, M., Harvey, B.P., Straub, S.C., Burrows, M.T., Alexander, L.V., Benthuysen, J.A., Donat, M.G. and Feng, M., 2019. Marine heatwaves threaten global biodiversity and the provision of ecosystem services. *Nature Climate Change*, *9*(4), pp.306-312.

Q31: Table 3 Time has units I guess (days). There is a misplaced bracket.

A31: Fixed.

Q32: L218 What's the empty bullet?

A32: Fixed.

Q33: L235 In this part or before you should provide references to earlier works on sea temperature extremes in the area, if any.

A33: We added some citations here, as three works on sea temperature extremes in the study area in the third paragraph of section 1 Introduction. Three references are Benthuysen et al., 2014, Feng et al., 2013 and Feng et al., 2021.

References:
Benthuysen, J., Feng, M., and Zhong, L.: Spatial patterns of warming off Western Australia during the 2011 Ningaloo Niño: Quantifying impacts of remote and local forcing, Continental Shelf Research, 91, 232-246, https://doi.org/10.1016/j.csr.2014.09.014, 2014.
Feng, M., McPhaden, M. J., Xie, S.-P., and Hafner, J.: La Niña forces unprecedented Leeuwin Current warming in 2011, Scientific Reports, 3, 1277, https://doi.org/10.1038/srep01277, 2013.
Feng, M., Caputi, N., Chandrapavan, A., Chen, M., Hart, A., and Kangas, M.: Multi-year marine cold-spells off the west coast of Australia and effects on fisheries, Journal of Marine Systems, 214, 103473, https://doi.org/10.1016/j.jmarsys.2020.103473, 2021.

Q34: L240 is it reasonable to assign 0 degrees Celsius in the water column?

A34: This is a pre-process training in SOM, not a final step to fill gaps. In the pre-process training, NaN values are assigned to zeros values so that SOM can create, initialize, and train the maps as requirements of the SOM package.

Q35: L246 I am confused, aren't you using ITCOMPSOM as stated earlier?

A35. We used the original SOM method, not ITCOMPSOM. We remove the redundant words in section 2.2 about the SOM method.

Q36: L325 what is '2015' here?

A36: Fixed.

**Reviewer #3:** Giuseppe M.R. Manzella

We want to thank the reviewer's constructive comments.

Q1: There are always essential elements to consider, but which do not seem to be well clarified in the article: how much does the non-linearity and variability of phenomena in the coastal area weigh on the method?

Figures 3 and 4 should be discussed on the basis of point 1. Before even getting to them I was in fact convinced that the method worked well with parameters such as temperature (or even salinity) but would have had significant errors with velocities.

A1: We appreciate this comment. SOM is an unsupervised learning method that is capable of capturing nonlinear processes in the training data. However, as a statistical method, it relies on enough realizations in the training dataset to properly capture the nonlinearity. Liu and Weisberg (2005) showed that the SOM method, unlike the linear EOF, was able to reveal asymmetric features in the Florida Current system, such as variations in current strength and coastal jet location. Many factors contribute to the non-linear variability in both temperatures and velocities on the Rottnest shelf. Mesoscale eddy can stem from the instability of the Leeuwin Current. Intense land-sea breezes during the summer drive amplification of near-inertial currents (Mihanovic, 2016). Additionally, the strong shear zone between the Capes Current and the Leeuwin Current during summer as well as interactions between the strengthening of the LC and the Perth Canyon in winter, may generate sub-mesoscale eddies. Due to their randomness, these non-linear processes may not be well captured in the daily temperature and velocity training datasets. This is especially true for the current velocity.

 In line with our findings, Sloyan et al. (2023) observed that although the R-squared for filled temperatures is nearly 1, the values for velocities were lower (0.7 for zonal velocity and 0.8 for meridional velocity, as shown in their Figures 6 and 8).

We added one paragraph in Section 6 to discuss this comment.

*"SOM is an unsupervised learning method capable of capturing non-linear processes in the training data. However, as a statistical approach, it relies on enough realizations in the training dataset to properly capture the nonlinearity.  showed that the SOM method, unlike the linear empirical orthogonal functions (EOF), was able to reveal asymmetric features in the Florida Current system, such as variations in current strength and coastal jet location. In the Rottnest shelf region, several factors contribute to the non-linear variability in both temperature and velocity fields.  Intense land-sea breezes during summer amplify near-inertial currents (Mihanovic et al., 2016). Additionally, the strong shear zone between the Capes Current and the Leeuwin Current in summer, as well as interactions between the strengthening the LC and the Perth Canyon in winter,*

*can generate sub-mesoscale eddies (Cosoli et al., 2020). SOM may well capture the mesoscale processes in the LC. Due to their randomness, however, submesoscale processes may not be fully captured in daily velocities. This is reflected in the lower R-squared values for velocities compared to temperatures (Figs 3, 4)."*

References:

Cosoli, S., Pattiaratchi, C., and Hetzel, Y.: High-frequency radar observations of surface circulation features along the south-western australian coast, Journal of Marine Science and Engineering, 8, 97, ARTN 97
10.3390/jmse8020097, 2020.

Mihanovic, H.; Pattiaratchi, C.B.; Verspecht, F. Diurnal sea breezes force near-inertial waves along Rottnest continental shelf, Southwestern Australia. J. Phys. Oceanogr. 2016, 46, 3487–3508.

Liu, Y. and Weisberg, R. H.: Patterns of ocean current variability on the West Florida Shelf using the self-organizing map, Journal of Geophysical Research: Oceans, 110, Artn C06003
10.1029/2004jc002786, 2005.

Sloyan, B. M., Chapman, C. C., Cowley, R., and Charantonis, A. A.: Application of Machine Learning Techniques to Ocean Mooring Time Series Data, Journal of Atmospheric and Oceanic Technology, 40, 241-260, https://doi.org/10.1175/Jtech-D-21-0183.1, 2023.

Q2: Are the data sufficiently representative of the physical state of the sea ? (perhaps the answers are in the articles cited by the authors, but a brief summary would have been very useful).

A2: The dataset used here is sufficient to capture the dominant alongshore and cross-shore processes on the Rottnest Shelf from intraseasonal to interannual time scales (Feng et al., 2024 and relevant studies cited in this manuscript, such as Feng et al. 2013; Benthuysen et al. 2014). The mooring observations capture the Leeuwin Current variability on the shelf. The data also has a good representation of the Capes Current, driven by strong southerly winds, flows northward, primarily confined to the middle shelf (20-50 m).

The mooring array does not well capture the sub-mesoscale processes on the shelf and those associated with Perth Canyon. These processes would cause errors in the SOM calculations and gap-filling, resulting in high uncertainty in the mapped velocity fields. Section 1 provides a brief overview of the dynamics in the study area. We also added some clarification in the abstract.

References:

Ming Feng, Toan Bui, Jessica Benthuysen, 2024. Seasonal climatology of the Leeuwin Current – Capes Current system derived from moored observations off southwest Australia. Journal of Geophysical Research: Oceans (Under review).

Q3: Line 77. Satellite data are used to extend the temperature to the surface. Since these data are part of sea truth exercises, a very brief presentation of associated precisions and uncertainties would be useful.

A3: We added the following sentences in section 2.1.1 Temperature to clarify this comment:

*"For data completion, we use satellite sea surface temperature (SST) sourced from the Regional Australian Multi-Sensor SST Analysis (RAMSSA) version 1.0 (Beggs et al., 2011), to extend the temperature data at each mooring to the sea surface by linear interpolation. The RAMSSA system combines SST data from infrared and microwave sensors on polar-orbiting satellites with in situ measurements to generate daily foundation SST estimates. North of 40°S RAMSSA is on average within ±0.07 °C of other multi-sensor SST analyses (Beggs et al., 2011). From conductivity-temperature-depth (CTD) profiles in the study region, ocean temperatures vary mostly linearly in the near-surface layer (top 30 m, below the foundation SST depth), so linear interpolation is an acceptable approximation."*

References:

Beggs, H., Zhong, A., Warren, G., Alves, O., Brassington, G., and Pugh, T.: RAMSSA—An operational, high-resolution, regional Australian multi-sensor sea surface temperature analysis over the Australian region, Australian Meteorological and Oceanographic Journal, 61, 1, 2011.

Q4: Line 78. The temperature in each mooring is extended to the surface with a linear interpolation. No problem with the seasonal thermocline?

A4: We inspected some CTD profiles in the study area, and we found that the ocean temperatures vary mostly linearly in the near-surface layer (top 30 m, below the foundation SST depth), so linear interpolation is an acceptable approximation. Therefore seasonal thermocline may not pose a significant issue.

Q5: The data are interesting and should be published. But I agree with one of the other referee: possible applications of gap filled data should be discussed, not only on heat waves.

A5: This is addressed in the reply to Q3 from Reviewer#2.

---

## Author Response (AR2)

**Author Response 2:**

**Gap-filled subsurface mooring dataset off Western Australia during 2010–2023**

Toan Bui[1], Ming Feng[1], Christopher Chapman[2]

[1]CSIRO Environment, Indian Ocean Marine Research Centre, Crawley, WA, Australia

**[2]**CSIRO Environment, Hobart, Tasmania, Australia

*Correspondence to:* ming.feng@csiro.au

We would like to thank the reviewer's constructive comments. Below, we provide our detailed point-by-point responses to the review comments. The reviewer's comments are in black, our response is in regular blue colour, and our revisions in the manuscript are in ***bold orange italics***.

Reviewer #2: Anonymous Referee

Q1: The authors addressed the concerns raised on the previous version. The number of figures in the Supplement is now 12, more than in the main document. I suggest to consider if really all of them are needed or they can be rearranged (e.g. merge S4/5). Moreover, for all line plots I would avoid using green/red combinations; blue/red as used elsewhere is preferable.

A1: The Supplement had 14 figures. We merged Fig. S4/5 and removed Fig. S13, so there are now 12 figures. We changed to blue/red combinations for all line plots in the Supplement and manuscript. We also updated the colour schemes, which allow readers with colour vision deficiencies to interpret our results correctly.

Q2: I have checked the netCDF file metadata, and variables latitude is misspelled. Authors may want to double check this and add an errata file on the website.

A2: Thank you for your comment. We added an errata file on the website (https://doi.org/10.25919/myac-yx60).

Q3: L117 This sounds repetitive.

A3: We rephrased the text to avoid the repetition.

*" The velocity observations on the IMOS mooring array are recorded by various RDI and Nortek ADCP instruments, typically sampling at 15-minute intervals, mounted in an upward-looking configuration above the seabed (Table 1)."*

Q4: Fig 2 caption: referent or reference?

A4: We updated the text, and only used the term "referent" in this caption.

Q5: Table 2 the crosses should perhaps be substituted by N/A or similar.

A5: Fixed.

Q6: Eq 2 the square brackets seem unnecessary; however you should add the bounds of the summation for clarity.

A6: In section 2.2 SOM method, we updated Eq 1 and 2 as follows:

*"Firstly, we estimated the local correlations in the data space, represented by a $cor_{i,j}^u$ matrix.*

$$cor_{i,j}^u = 1 + \sqrt{\sum DAT\_cor^2}, \qquad\qquad (1)$$

*where DAT_cor is a correlation matrix among each normalized input vectors within a SOM unit; $cor_{i,j}^u$ is the local correlation matrix between the missing and the mean of all the observed training data within the SOM unit u.*

*Given with local correlations in the data space, we then calculated the minimum Euclidean distance between a normalized input vector X containing missing and non-missing components and the referent vector of the SOM unit, $ref^u$ using a similarity function (Chapman and Charantonis, 2017). The similarity function is defined as:*

$$sim(X, ref^u) = \sum_{i \in non-missing} \left(1 + \sum_{j \in missing} (cor_{i,j}^u)^2\right) \times \sqrt{(X_i - ref_i^u)^2}, \qquad (2)$$

*Where $X_i$ is the non-missing data in X, $ref_i^u$ is the mean of all training data in the SOM unit u.*

Reference:

Chapman, C. and Charantonis, A. A.: Reconstruction of subsurface velocities from satellite observations using iterative self-organizing maps, IEEE Geoscience and Remote Sensing Letters, 14, 617-620, https://doi.org/10.1109/Lgrs.2017.2665603, 2017.

Q7: Fig S7: the caption, especially the last part, reads repetitive.

A7: We kept the last part because we wanted to clarify the climatology estimate, and we do not have this sentence in the manuscript.

Q8: Fig S9: I would avoid adding comments/conclusions in the captions, these should be in the main text.

A8: We removed the conclusions in the captions.

Q9: I suggest to perform some formatting changes but I do not have other major comments, except perhaps stressing in the text the limitations of the SOM-derived product (some systematic biases may exist, see comments below).

L210 I have the impression the SOM tends to have some warm bias, e.g. the extent of extremes is smaller. This should be discussed.

A9: We added one paragraph in Section 6 Summary and discussion to address these comments.

*"There are weak biases in the SOM-derived product, such as a warm bias at WATR20 during the validation period from January 1 to May 30, 2020 (Fig. S6c). It is noted that this period experienced multiple marine cold spells (Fig. 7b). This systematic bias is likely due to the nature of the SOM algorithm, which tends to underestimate the magnitude of extreme events while effectively capturing broader patterns. Future work could explore bias correction techniques to enhance accuracy."*

Q10: L251 have you introduced Categories already?

A10: We added two sentences in the last paragraph of Section 3 Data application to introduce Categories.

*"In this study, different categories of MHWs are defined based on multiples of the local difference between the climatological mean and the $90^{th}$ percentile (Hobday et al., 2018). The magnitude scale descriptors classify MHWs as moderate (between 1–2 multiples, Category I), strong (2–3, Category II), severe (3-4, Category III), and extreme (>4, Category IV)."*

Reference:

Hobday, A. J., Oliver, E. C., Gupta, A. S., Benthuysen, J. A., Burrows, M. T., Donat, M. G., Holbrook, N. J., Moore, P. J., Thomsen, M. S., and Wernberg, T.: Categorizing and naming marine heatwaves, Oceanography, 31, 162-173, 2018.

Q11: L332/3 These very short sentences are strange; is something missing?

A11: This is the journal's standard format. Nothing is missing.